# Neighborhood Learning in Weighted Beeping Networks

## Abstract

Neighborhood Learning (NL) is a fundamental tool in Multiagent Systems (MAS). The task is for each autonomous agent to learn, in parallel, some information (e.g., agent identifier, message, etc.) from every neighboring agent, according to some notion of vicinity. NL thus requires communication among neighboring agents, which is particularly challenging if agents are tiny devices with very limited capabilities (for instance, biological systems) and may interrupt each other. In this work, we study how the speed of learning depends on the system topology. We model the communication environment as a Weighted Beeping Network (WBN). In a WBN, network nodes (one for each agent) communicate by deciding whether to beep or stay silent – all the beeps are then scaled by weights on the corresponding links, and a threshold function is applied at each idle node to check if they heard a beep or not. We introduce a novel characteristic of a WBN topology, called Maximum Average Influence (MaxAveInf), and we prove almost tight upper and lower bounds on the running time to accomplish NL task by a Multiagent System, as a linear function of that characteristic. Although MaxAveInf is a global characteristic and it could be as large as the number of all agents in some networks, even with small neighborhoods, for networks with small value of MaxAveInf we succeeded to give a provably-efficient nearly-optimal algorithmic solution.

## 1 Introduction

In this work, we study Neighborhood Learning (NL) in Multiagent Systems (MAS). To solve NL, each agent must, in parallel, receive an information from every neighboring agent, according to some notion of vicinity[1] (see Section 2 for a formal definition of NL). Parallel NL is then a fundamental tool for typical computations in MAS where agents must communicate with all neighbors (e.g., Ba et al. (2024); Chen et al. (2024a); Du et al. (2024); Li et al. (2024); Liu et al. (2024b)). (See Section A.3 in the Appendix for details on other applications.)

Smart agents' systems are common in nature, and their environment is often harsh and communication-restrictive, e.g., insect colonies, bird flocks, bacteria populations, and others (see the recent survey by Liu et al. (2024a)). As a subfield of *Swarm Intelligence*, smart agents enable a collective of organisms or entities to exhibit a whole-system intelligence. This is done by having each member communicate and analyze (typically) simple "wireless" signals between peers.

One of the simplest classic models of ad hoc communication reflecting those natural mechanisms is the Beeping Network (BN) Cornejo & Kuhn (2010); Afek et al. (2011; 2013). BNs are graphs where a pair of nodes are neighbors if they can communicate directly (i.e., without relays), modeled with a link between them. At any given time a node can either *beep* (emit some signal with no embedded information but the presence of the signal) or *listen*. The core limitation of the standard BN model is in its simplistic handling of interference: a listener cannot distinguish between a single beep

---

[1]Notice that complex learning tasks can be implemented using a Neighborhood Learning algorithm (see Section A.3 in the Appendix), as long as it facilitates the exchange of arbitrary long messages (not only ID's) among agents, but the problem is called Neighborhood Learning only to convey that agents do not need to know initially the other agents they can communicate with. Also, we define NL based on the scenario where nodes receive information from all neighbors as a worst-case definition designed to ensure maximum versatility and robustness. This comprehensive definition does not imply that full communication is a prerequisite or necessity for every specific application.

and multiple simultaneous beeps. This fails to model real-world phenomena where the perceived strength of a signal (interference, environmental features) varies significantly.

In this work, we extend the BN communication model to weighted graphs, which we call Weighted Beeping Networks (WBN). The WBN model addresses BN limitations by:

- Introducing weighted links to model arbitrary influence of a beeper on a listener.
- Defining a precise Delivery Condition: a beep is successfully received only if the aggregated influence of all other simultaneously beeping nodes is below a threshold (normalized to 1).

More specifically, in WBN's an ordered pair of nodes is connected by a link if the weight associated with that link is positive; here, weights on links model various environmental features that *influence* the perceived strength of a beep. Notice that this modeling choice does not restrict the scope of application of our study, since a non-existing link is simply a link with weight zero, and nodes do not know link weights. Then, a beep from a *beeper* $u$ is received by a listener $v$ depending on the influence of $u$ on $v$ (i.e., the weight of link $(u, v)$) with respect to the influence of other beepers. Accordingly, we define the NL problem restricted to *influential* neighbors, by setting a *parametric threshold* $\sigma$ for a neighbor to be influential that depends on the application.[2] The WBN model is a simple (weighted-graph based) yet fundamental (threshold for delivery of weighted information, every info bit accounted for by some beep) MAS model. Many "richer" wireless communication models, e.g., Radio Networks and SINR Kowalski et al. (2020); Kowalski & Mosteiro (2025), can be implemented in WBN, whereas biological communication systems, e.g., bacteria signaling Moreno-Gámez et al. (2023) can be abstracted by WBN using binary coding for different communication molecules and specific threshold for combined influence (see Appendix A.2).

**Our contribution and broader discussion.**

- Modeling Question: we introduce a *Maximum Average Influence* ($W$) as the fundamental topological characteristic governing complexity.
- Algorithmic Question: we present a provably efficient Neighborhood Learning (NL) algorithm whose running time is nearly-optimal in terms of $W$.
- Complexity Question: we prove the almost tight $\Omega(W)$ lower bound, showing that parallel learning can be costly even in sparse networks with high $W$.

More specifically, we study Neighborhood Learning in a MAS of $n$ agents under the WBN model, where each agent is installed in one network node. In particular, we show how to design a provably efficient NL algorithm, run autonomously in parallel by agents. In this work, we assume that nodes communicate synchronously in *rounds*, and we measure algorithmic performance in rounds of communication.

Most importantly, we identify a characteristic of WBNs, which we call Maximum Average Influence (MaxAveInf) and denote as $W$. The NL problem, similarly to almost all complex graph problems, does not have an exact formula for the number of rounds needed to solve the problem, only an approximation as a function of some graph characteristic that holds asymptotically. E.g., time-related parameters (mixing, covering, etc.) of random walks are only approximated by concise formulas, such as vertex/edge expansion or conductance. We follow the same methodology and show mathematically that

> $W$ *is a good approximation of the minimum number of rounds needed to solve the NL problem in parallel*[3],

by showing that the ratio between the upper and lower bound is $O(\log^2 n)$. More precisely, we show in Sec. 3 that our NL algorithm locally propagates messages of length $\mathcal{M}$ in $O(W \log n \cdot (\mathcal{M} + \log n))$ rounds on *any* topology graph $G$ with MaxAveInf at most $W$ (see Theorem 2). (Note that the additive $\log n$ comes from attaching the node identifier to the message, and it subsumes $\mathcal{M}$ for short messages, i.e., $\mathcal{M} = O(\log n)$.) This bound holds with high probability, i.e., with error polynomially small in $n$. For clarity, we present and analyze our algorithm in three steps: first we present a simplified version of NL, called *One-beep Local Broadcast (OBLB)*, in which only one bit of information has to be delivered (i.e., $\mathcal{M} = 1$) – for settings where only $W$ and a polynomial upper

---

[2]This threshold models a fundamental physical reality: signal detection requires sufficient signal-to-noise ratio. Our specific threshold is normalized to 1, but it can be generalized to any constant.

[3]Recall that to solve NL each agent must receive an information from every neighboring agent.

bound on $n$ are known, and all agents start execution simultaneously (Sec. 3.1); then we generalize our techniques to NL (Sec. 3.2); and finally, we show how to remove the knowledge of $W$ and synchronization assumptions, and discuss the case of dynamic weights (Sec. 3.3 and App. C).

In Section 4, for OBLB we prove an almost matching lower bound of $\Omega(W)$ (see Theorem 3). This lower bound automatically holds for NL, as a more general problem. To prove the lower bound, we show an adversarial input on which any OBLB algorithm requires the claimed time. Motivated by the topology of classic neural networks, we used a bipartite graph with certain weight function – we conjecture that our approach could be further extended to other more specific network classes, such as multi-layered networks[4] or scale-free graphs. In fact, our extensive experimental study strongly indicates that such lower bound applies to a variety of real-world and (commonly used) synthetic topologies (e.g., grids or scale-free graphs).

Our technical results above have important implications on the efficiency of parallelization of Neighborhood Learning. MaxAveInf is a global characteristic that could be as large as the number of all agents, even if the influential neighborhoods (i.e., the sets of influential links connected to each receiver) are small.[5] Given the locality of the NL problem, one would expect to solve it fast in parallel in small neighborhoods. However, our lower bound shows that executing NL in parallel may be costly, even in networks with small degree (but with high MaxAveInf). To the best of our knowledge, ours is the first theoretical analysis of upper and lower bounds for the NL in WBN model.

Additionally, in Appendix E, we include a thorough experimental evaluation of our algorithms on two potential learning environments: social networks in the natural world (four datasets), on top of which nano-devices could be deployed, and ad-hoc infrastructures (3D-grid, scale-free nets), deployed possibly for other purposes, for instance, for simultaneous scanning and learning. Our experiments show that our theoretical analysis is almost tight and constants hidden in the asymptotic notation are reasonably small. We also show that performance of our algorithm is much better than other algorithms (such as round-robin or periodic transmissions schedules with initial random shift).

## 2 MODEL AND DEFINITIONS

### 2.1 MULTIAGENT SYSTEM MODEL

We consider a Multiagent System (MAS) formed by $n$ autonomous agents, each installed on a node of a Weighted Beeping Network (WBN). Nodes have communication and computational capabilities, and each node is labeled with a unique ID of $O(\log n)$ bits. Time is discretized in **slots**, also called **rounds**. In each slot a node may be either **beeping**, i.e., sending a **beep**,[6] or **listening**. A time slot is long enough to beep, or hear a beep if listening (see Sec. 2.2 for details), and possibly perform one bit read/write/compare operation in local memory, which are assumed to take negligible time with respect to the time required by the communication. When presenting main ideas, we assume clock synchronization, but we also show how to get rid of this limitation later (Sec. 3.3 and App. C).

### 2.2 COMMUNICATION MODEL ON GENERAL BIDIRECTED WEIGHTED GRAPHS

Due to various reasons (communication interference, preference, etc.), the beep of a node may be perceived differently by two other nodes. We model this communication phenomenon with an **influence** function $w : V \times V \rightarrow \mathbb{R}^+ \cup \{0\}$ where $V$ is the set of network nodes, and for any $x, y \in V$, $w(x, x) = 0$ and $w(x, y) = w(y, x)$. The network topology is modeled by a *general* bidirected weighted graph $G = (V, E)$, where $E = \{x, y | x, y \in V \wedge w(x, y) > 0\}$ is the set of bidirected links,[7] and the weight function is the influence function defined above. An example of a topology graph can be seen in Figure 1a (at the beginning of the Appendix). A link from **beeper** $x \in V$ to **listener** $y \in V$ is denoted as $(x, y)$, and the set of nodes beeping at time $t$ is denoted as $V(t)$. In each time slot, a listening node may **hear**: **silence** if none of its neighbors in $G$ beeps in this time slot, or a **beep** otherwise.

---

[4]Multi-layered WBNs could be seen as beeping neural networks, similarly to spiking neural networks.

[5]E.g., each node has only one influential link, and all remaining links have influence just below the threshold.

[6]Some kind of a signal, such as radio, light, sound, etc.

[7]Even though the weight is the same in both directions, we use bidirected links to be able to specify the direction of communication when needed. Graph $G$ can be *arbitrary* – weight 0 means no edge in $E$ (no influence).

## 2.3 Neighborhood Learning and Simplified OBLB Problem

We start with the following definitions. For any constant $\sigma \geq 1$, called **influence threshold**, let $E_\sigma \subseteq E$ be the subset of links with influence at least $\sigma$, that is $E_\sigma = \{(x,y) \in E | w(x,y) \geq \sigma\}$. We call these links **influential**.[8] A **message** is a sequence of bits held by a node for the purpose of sending it to other node(s). We assume that each node has a unique identifier, also a sequence of bits, called **ID**. We say that a node **analyzes** a sequence of beep/silence heard when it decodes such sequence into a sequence of bits, according to a previously established coding system. When a node $x$ receives some information from some other node $y$ after analyzing the sequence of beep/silence heard, we say that $x$ has **learned** the information sent by $y$.

**Definition 1.** *Given a constant $\sigma \geq 1$ and a WBN with set of influential links $E_\sigma$ where each node holds a message, the **Neighborhood Learning Problem** is solved in $\Gamma$ time slots if, for each influential link $(x,y) \in E_\sigma$, by the end of slot $\Gamma$ node $y$ has learned the ID and message of node $x$ by analyzing the sequence of beep/silence heard.*

The threshold $\sigma$ restricts learning only to neighbors connected by links in $E_\sigma$. Nevertheless, nodes with lower influence are still around and may interfere with the reception of information from influential neighbors (see the delivery condition below).

In order to design and analyze efficient algorithms for Neighborhood Learning, we introduce a simpler problem. The main concept here is a "delivery of a beep" from $x$ to $y$ (with $w(x,y) > \sigma$), which describes a situation when node $y$ heard a beep while node $x$ beeped, but also that the influence of the other beepers on $y$ was small. More formally, we say that a beep from node $x$ was **delivered** to node $y$ in some time slot $t$, if the following happened in time slot $t$:

> **(Delivery Condition:)** $x$ beeped, $y$ listened, and the aggregated influence on $y$ of all other beepers in $V(t)$ was less than 1 (i.e. $\sum_{z \in V(t): z \neq x} w(z,y) < 1$).

Figure 1b (at the beginning of the Appendix) illustrates an example of delivery. Note that $y$ may not recognize whether the heard beep was actually a delivery of a beep from $x$, yet assuring that such deliveries occur eventually is an important intermediate goal to achieve.

The simplified version of the main problem is as follows:

**Definition 2.** *Given a constant $\sigma \geq 1$ and a WBN with set of influential links $E_\sigma$, the **One-beep Local Broadcast (OBLB) Problem** is solved in $\Gamma$ time slots if, for each influential link $(x,y) \in E_\sigma$, by the end of slot $\Gamma$ a beep has been delivered from $x$ to $y$.*

## 2.4 MaxAveInf Metric Function of Algorithms

We measure algorithmic performance in time slots as a function of $|E_\sigma|$ and the following characterization function of a WBN, called **Maximum Average Influence** (**MaxAveInf**):

$$W = \max_{E' \subseteq E_\sigma} \frac{1}{|E'|} \sum_{(x,y) \in E'} \sum_{(z,y) \in E} w(z,y).$$

MaxAveInf is the maximum cumulative influence the receiver of an average link in any subset of $E_\sigma$ (links with influence above threshold) can experience, from the transmitters of all other links. Increasing the parameter $\sigma$ we restrict the information to be learned (neighborhoods may shrink), but we may learn it faster as $W$ is monotonically non-increasing with growing $\sigma$. In Figure 1 (at the beginning of the Appendix) for example, setting $\sigma = 1$ means that only 4 links have to deliver information (beeps). Nevertheless, links with smaller influence may still deliver some "false beeps" that the algorithm has to mitigate.

The analysis of our Neighborhood Learning algorithms holds **with high probability (whp)**, i.e., with probability at least $1 - n^c$ for some arbitrary constant $c > 0$. This constant could scale up by increasing the constant factor embedded in the asymptotic formula (on the number of beeping rounds).

---

[8]Influence is defined as a link attribute because a given beeper may have different influence on different neighboring listeners. Nevertheless, given that influence is a concept of beepers on listeners, we also refer to the influence of (or on) nodes for clarity of exposition.

# 3 Algorithmic Upper Bound

We first present and analyze a solution to the simplified one-beep version of NL, called One-beep Local Broadcast as in Def. 2 (Sec. 3.1). Next, we generalize it to any NL input (Sec. 3.2). Extensions of our algorithms are given in Sec. 3.3, and full versions of sketched proofs are in the Appendix.

## 3.1 One-beep Local Broadcast

Consider the following One-beep Local Broadcast algorithm:

*In each time slot, each node beeps with probability $p \leq \frac{1}{4W}$ .*

The following analysis shows correctness and upper bound on the running time of this algorithm.

**Lemma 1.** *Consider any subset of links $E'_\sigma \subseteq E_\sigma$, such that for each $(x, y) \in E'_\sigma$ node $x$ beeps with probability $p \leq 1/(4W)$ and the remaining nodes are idle. Then, the expected number of links in $E'_\sigma$ where a beep is delivered is at least $|E'_\sigma|(1 - p)p/4$.*

*Proof sketch:* For a given local broadcast protocol, let $X_x(t)$ be a random variable indicating whether node $x \in V$ beeps at time $t$, and let $X_{x,y}(t)$ be a random variable indicating whether a beep from $x$ was delivered to $y$ at time $t$. To prove the lemma, we need to show: $\mathbf{E}\left(\sum_{(x,y) \in E'_\sigma} X_{x,y}(t)\right) \geq |E'_\sigma|(1 - p)p/4$.

By linearity of expectation, for $M = \{(x, y) \in E'_\sigma | \sum_{z \in V(t)} w(z, y) \leq 2W\}$ :

$$\mathbf{E}\Big(\sum_{(x,y) \in E'_\sigma} X_{x,y}(t)\Big) = \sum_{(x,y) \in M} \mathbf{E}\left(X_{x,y}(t)\right) + \sum_{(x,y) \in E'_\sigma \setminus M} \mathbf{E}\left(X_{x,y}(t)\right) \geq \sum_{(x,y) \in M} \mathbf{E}\left(X_{x,y}(t)\right)$$

$$= \sum_{(x,y) \in M} Pr(X_{x,y}(t) = 1) = \sum_{(x,y) \in M} Pr(X_{x,y}(t) = 1 | X_x(t) = 1 \wedge X_y(t) = 0) \cdot p(1 - p) .$$

Recall that, for any $(x, y) \in E$, the conditions to deliver a beep from node $x$ to node $y$ at time $t$ are: $X_y(t) = 0$, and $w(x, y) \cdot X_x(t) \geq \sigma$, and $\sum_{z \in V(t):z \neq x} w(z, y)X_z(t) < 1$. From the conditions to deliver a beep and by Markov inequality,[9] we bound complementary event:

$$Pr\Big(X_{x,y}(t) = 0 | X_x(t) = 1 \wedge X_y(t) = 0\Big) = Pr\Big(\sum_{z \in V(t):z \neq x} w(z, y)X_z(t) \geq 1\Big) \leq p \sum_{z \in V(t)} w(z, y) .$$

By upper bounding $\sum_{z \in V(t)} w(z, y) \leq 2W$ (by definition of $M$) and $p \leq 1/(4W)$ (by the condition of the lemma), we get $Pr\left(X_{x,y}(t) = 0 | X_x(t) = 1 \wedge X_y(t) = 0\right) \leq 1/2$. Putting them all together, we have that $\mathbf{E}\left(\sum_{(x,y) \in E'_\sigma} X_{x,y}(t)\right) \geq \sum_{(x,y) \in M} \left(\frac{1}{2} \cdot p(1 - p)\right) \geq |M|p(1 - p)/2$.

To conclude the proof, it remains to show that $|M| \geq |E'_\sigma|/2$. It follows from the Pigeonhole principle applied to $W = \max_{E' \subseteq E_\sigma} \frac{1}{|E'|} \sum_{(x,y) \in E'} \sum_{z \in V(t)} w(z, y)$, in which the complementary set $E'_\sigma \setminus M = \{(x, y) \in E'_\sigma | \sum_{z \in V(t)} w(z, y) > 2W\}$ must have at most $|E'_\sigma|/2$ elements. $\square$

**Theorem 1.** *For $p \leq \frac{1}{4W}$, One-beep Local Broadcast can be solved in $O(W \log |E_\sigma|)$ time slots whp.*

*Proof.* Consider an execution of a One-beep Local Broadcast protocol. Let $E_\sigma^{(t)} \subseteq E_\sigma$ be the set of links with influence at least $\sigma$ where a beep was not delivered before the beginning of time slot $t = 1, 2, \ldots$. That is, $E_\sigma^{(1)} = E_\sigma$. By Lemma 1, for $p \leq 1/(4W)$ we have that for any $t \geq 1$ it is

$$\mathbf{E}\left(|E_\sigma^{t+1}||E_\sigma^t| = i\right) \leq (1 - (1 - p)p/4)i .$$

---

[9]Note that Markov inequality does not require independence of the random variables (see Thm. 3.1 in Mitzenmacher & Upfal (2005)).

---

**Algorithm 1:** Neighborhood Learning algorithm, pseudo-code for a node $v$

---

**Input** (upper bounds on) parameters $n, W, \mathcal{M}$

$\tau \leftarrow$ upper bound on time computed locally in analysis of Theorem 1, based on given $n, W$

$\alpha_v \leftarrow$ beeping pattern computed according to the definition of $\alpha_v$ in the main text

**for** $i = 1, 2, \ldots, \tau$ **do**

    **if** $Random(1/(4W)) = 1$ **then**

        node $v$ beeps in the $2 \log n + \mathcal{M}$ rounds of the super-round $i$ according to pattern $\alpha_v$

    **else**

        node $v$ stays silent and records the feedback in the $2 \log n + \mathcal{M}$ rounds of the

        super-round $i$ into string $\beta_{v,i}$

**for** $i = 1, 2, \ldots, \tau$ **do**

    **if** $\beta_{v,i}$ *is valid* **then**

        node $v$ locally (no beeping round is needed) decodes $\beta_{v,i}$ into a pair of ID and message

**Output** decoded pairs (without redundancies), each containing node ID and message

---

Then, it is: $\quad \mathbf{E}\left(|E_\sigma^{(t+1)}|\right) \leq \sum_{i=0}^{|E_\sigma|} Pr(|E_\sigma^{(t)}| = i)(1 - (1-p)p/4)i$

$$= (1 - (1-p)p/4)\mathbf{E}\left(|E_\sigma^{(t)}|\right) = (1 - (1-p)p/4)^t |E_\sigma| \ .$$

Notice for the latter inequality that transmitters do not need to switch off because in Lemma 1 the influence is added over all neighbors, regardless of whether a beep has been delivered already or not.

On the other hand, by Markov inequality, the probability that One-beep Local Broadcast has not been solved at the beginning of time slot $t + 1$ is

$$Pr\left(|E_\sigma^{(t+1)}| \geq 1\right) \leq \mathbf{E}\left(|E_\sigma^{(t+1)}|\right) \leq (1 - (1-p)p/4)^t |E_\sigma| \ .$$

Fixing $t = (4/((1-p)p)) \ln(n|E_\sigma|)$, which is in $O(W \log n)$, we have that

$$Pr\left(|E_\sigma^{(t+1)}| \geq 1\right) \leq \left(1 - \frac{(1-p)p}{4}\right)^{\frac{4}{(1-p)p} \ln(n|E_\sigma|)} |E_\sigma| \leq \frac{1}{e^{\ln(n|E_\sigma|)}}|E_\sigma| \ = \ 1/n \ .$$

Thus, the claim of the theorem follows. $\qquad\qquad\qquad\qquad\qquad\qquad\qquad\qquad\qquad\qquad\qquad\qquad\square$

### 3.2 Neighborhood Learning (NL)

In this section, we generalize the algorithm and analysis from Section 3.1 to Neighborhood Learning (Definition 1). It is run locally and autonomously by each node $v$. The input contains, apart of the identifier of node $v$ (given by a binary string of length $\log n$) and an input message: the number of nodes $n$, the maximum average influence $W$, and the maximum size of a message $\mathcal{M}$. In Section 3.3 we show how to get rid of assumptions of known $W$ and synchronization, and relaxing the requirement of knowing $n$ to only a polynomial upper bound on $n$.

**Algorithm's structure.** The pseudo-code of the Neighborhood Learning algorithm is given in Algorithm 1. The algorithm partitions rounds into consecutive windows of time, called ***super-rounds***, each of length $2 \log n + \mathcal{M}$, where $\mathcal{M}$ is the maximum length of a message sent. The number of super-rounds, denoted by $\tau$, is computed as in the time analysis of the One-beep Local Broadcast algorithm in Theorem 1. The algorithm also uses a random generator $Random(1/(4W))$ that produces 1 with probability $1/(4W)$, and 0 otherwise.

**Super-round** $i$. In the beginning of a super-round $i$, each node $v$ performs local action as in round $i$ of the One-beep Local Broadcast algorithm (see Section 3.1). It uses the random generator. If the decision is to beep, then it ***beeps*** in its $i$th super-round ***according to the following beeping pattern, denoted*** $\alpha_v$:

- if there is bit 1 in position $j$ of ID of node $v$, it beeps and stay silent in rounds $2j - 1$ and $2j$ of its super-round, resp.

- if there is bit 0 in position $j$ of ID of node $v$, it stay silent and beeps in rounds $2j - 1$ and $2j$ of its super-round, resp.

- in rounds $2\log n + 1, \ldots, 2\log n + \mathcal{M}$ of its super-round, node $v$ beeps according to the bit representation of its message (i.e., it beeps for bit 1 and stays silent for bit 0); if there are less than $\mathcal{M}$ bits in the message, it piggybacks the message with the 0's until the end of the super-round.

If, however, the decision is not to beep, node $v$ stays silent during the whole super-round $i$ and records the received sequence of beep/silence into a binary string $\beta_{v,i}$.

We call a super-round $i$ **valid** for node $v$ if $v$ has been listening during the super-round and its recorded feedback $\beta_{v,i}$ has the following property: the number of 1's in the first $2\log n$ positions of $\beta_{v,i}$ is exactly $\log n$.

**Final decoding.** At the end of Algorithm 1, node $v$ **_decodes_** each $\beta_{v,i}$ of a valid super-round $i$:

- if there are bits 10 at positions $2j - 1, 2j$, for $j \le \log n$, it decodes bit 1 and puts it at position $j$ of the decoded ID;

- if there are bits 01 at positions $2j - 1, 2j$, for $j \le \log n$, it decodes bit 0 and puts it at position $j$ of the decoded ID;

- in round $2\log n + j$, for $j \le \mathcal{M}$, node $v$ decodes 1 and puts it in position $j$ of the decoded message if it heard a beep, and decodes 0 if there was a silence.

Then, node $v$ puts pairs, containing decoded ID and message, to its output, removing redundancies.

**Algorithm analysis.** We call a neighbor of node $v$ **influential** if the link between them is influential.

**Lemma 2.** *If there is only one influential neighbor of a node $v$ choosing to beep in the considered super-round $i$, and the aggregated influence on $v$ from other neighbors in this super-round is smaller than 1, then node $v$ correctly decodes the other node's ID and message from sequence $\beta_{v,i}$ at the end of the algorithm.*

*Proof.* The crucial observation is that if there is only one influential neighbor of $v$ that chooses to beep in the super-round $i$, say node $v^*$, and the aggregated influence from other neighbors of $v$ is smaller than 1, then the operations of beeping according to pattern $\alpha_{v^*}$ is directly received by $v$, in the sense that $\beta_{v,i} = \alpha_{v^*}$. In such case, observe that decoding is the actual reversed operation to encoding the pattern (i.e., reversed to creating pattern $\alpha_{v^*}$), and in this sense can uniquely decode the ID of $v^*$ (based on the first $2\log n$ bits of sequence $\beta_{v,i}$) and the message of $v^*$ (based on the remaining $\mathcal{M}$ bits of the sequence). □

**Lemma 3.** *If two or more influential neighbors of a node $v$ beep in the beginning of a super-round $i$, then node $v$ ignores the information received in this super-round in sequence $\beta_{v,i}$.*

*Proof.* It follows from the fact that $v$ can recognize that at least two of its influential neighbors beeped, or that the aggregated influence from other neighbors is at least 1, by analyzing the first $2\log n$ rounds of this super-round. More precisely, among these bits, there would be more than $\log n$ beeps in these rounds, while in the case of one influential neighbor and aggregated influence of others smaller than 1 there should be exactly $\log n$ beeps in the received sequence $\beta_{v,i}$. □

**Theorem 2.** *For $p \le 1/(4W)$, Neighborhood Learning can be solved in $O(W\log n \cdot (\log n + \mathcal{M}))$ time slots whp.*

*Proof.* We combine Theorem 1 with Lemma 2 to conclude that for any node $v$ and any of its influential neighbors $v^*$, there is a super-round $i$ when $v^*$ is the only influential neighbor of $v$ that beeps, and the aggregated influence of other neighbors of $v$ is smaller than 1. Thus, the ID and message of $v^*$ are correctly decoded by $v$ at the end of the algorithm.

Lemma 3 implies that node $v$ correctly ignores information that could come from more than one influential neighbor or be "too noisy" by aggregated influence of others – node $v$ correctly recognize it as non-valid.

The number of rounds is $\tau \cdot (2\log n + \mathcal{M})$. By Theorem 1, which analysis is used to compute $\tau$, and by asymptotic $|E_\sigma| = \Theta(n)$, the total number of time slots is $O(W\log n \cdot (\log n + \mathcal{M}))$. $\quad\square$

## 3.3 EXTENSIONS AND DROPPING OFF SOME LIMITATIONS

Our main algorithm in Sec. 3.2 can be extended in the following ways – see details in Appendix C.

**Arbitrary (unknown) $W$ and only polynomial upper bound on $n$.** The algorithm only needs a polynomial upper bound on $n$, as $n$ occurs under logarithm in the formula on MaxAveInf in Theorem 2. The algorithm could be transformed into one without the need of knowing $W$ in advance, by doubling estimates of $W$, using acknowledgments and stopping condition. The number of steps increases only by a constant factor.

For the sake of our experiments in Appendix E, we designed a stochastic estimation of $W$, which is quite accurate on the tested inputs, without significant affectance on algorithm performance.

**De-synchronized systems.** Due to a relatively simple structure and various checking mechanisms in our algorithm, it can be extended to de-synchronized setting (with clock shifts) by using the known synchronizing methods for beeping networks, see e.g., the recent work by De Marco & Kowalski (2025). These methods require only a constant overhead, and transform algorithms with structures similar to ours from fully synchronized into de-synchronized solutions.

**Dynamic weights and noise.** Noise and other physical conditions are captured by link weights. Although NL cannot be solved in WBN under fully dynamic weights, it could if dynamicity is limited.

## 4 LOWER BOUND

In this section we prove that any Neighborhood Learning algorithm takes at least $\Omega(W)$ beeping rounds in some WBN. In fact, even a simplified One-beep version of the problem requires such number of beeping rounds in some networks.

In the analysis of this section we will show an adversarial WBN that requires the claimed running time. The network topology of such WBN is a complex bipartite graph (non-existing links correspond to weight $0$ between the end nodes). We start with a technical lemma (with detailed proof and additional figures in Appendix D), followed by the main theorem.

**Lemma 4.** *Consider a WBN with a complete bipartite topology graph $(B, L, E)$, where $B$ is a set of beepers, $L$ a set of listeners, $E = \{(b, \ell) | b \in B \wedge \ell \in L\}$, and $|B| = |L|$. The network is embedded in a metric space with distance function $d : B \times L \to \mathbb{R}^+$. The influence function is $w(b, \ell) = d_{\min}/d(b, \ell)$ for all $(b, \ell) \in E$, where $d_{\min}$ is the smallest length among all links in $E$.*

*Consider some link $(b', \ell') \in E$ and a set of links $\mathcal{E} \subset E$ such that $(b', \ell') \notin \mathcal{E}$. Then, if it is possible to deliver a beep in each link $(b, \ell) \in \mathcal{E}$ in one time slot, it is $\sum_{(b,\ell)\in\mathcal{E}} w(b, \ell') \in O(1)$.*

*Proof sketch:* To prove the claim we split the set of links $\mathcal{E}$ into *distant* and *close* with respect to listener $\ell'$. For the set $\mathcal{E}_{close}$ of close links, we upper bound the influence of their beepers on $\ell'$ simply by $|\mathcal{E}_{close}|$ (since each link has influence at most 1), and we bound $|\mathcal{E}_{close}|$ by a constant using a geometric argument and the fact that a beep is delivered through every link in $\mathcal{E}_{close}$. For the set $\mathcal{E}_{dist}$ of distant links, we bound the influence of their beepers on $\ell'$, by their influence on a listener at shortest distance from $\ell'$. We bound the latter by a constant using a geometric argument and the fact that a beep is delivered through every link in $\mathcal{E}$. The details follow.

Let $2r$ be the shortest distance from any listener in $\mathcal{E}$ to listener $\ell'$. Let $\mathcal{E}_{close} = \{(b, \ell) \in \mathcal{E} | d(b, \ell') \leq r\}$, that is the set of links $(b, \ell) \in \mathcal{E}$ whose beeper $b$ is within distance $r$ from listener $\ell'$. We bound first the influence of close beepers on $\ell'$ as

$$\sum_{(b,\ell)\in\mathcal{E}_{close}} w(b, \ell') = \sum_{(b,\ell)\in\mathcal{E}_{close}} \frac{d_{\min}}{d(b, \ell')} \leq |\mathcal{E}_{close}| < 4 . \tag{1}$$

The latter inequality can be proved using a geometric argument (whose boundary case is illustrated in Figure 3a in the Appendix), and the conditions for successful delivery of a beep.

We now bound the influence of beepers in $\mathcal{E}_{dist} = \mathcal{E} \setminus \mathcal{E}_{close}$ on listener $\ell'$ as follows.

$$\sum_{(b,\ell) \in \mathcal{E}_{dist}} w(b, \ell') = \sum_{(b,\ell) \in \mathcal{E}_{dist}} \frac{d_{\min}}{d(b, \ell')} < 6 \,, \tag{2}$$

where the latter inequality can be also proved using a geometric argument (whose boundary case is illustrated in Figure 3b in the Appendix), and the conditions for successful delivery of a beep. Combining the bounds in Equations 1 and 2, the claim follows. □

**Theorem 3.** *There exists a WBN and influence function such that, to solve the Neighborhood Learning problem, $\Omega(W)$ time slots are required with probability $1$, where $W$ is the MaxAveInf.*

*Proof.* We prove the claim showing an adversarial WBN where even One-beep Local Broadcast in a subset of links requires the claimed running time. Specifically, consider a complete bipartite WBN $(B, L, E)$ as required by Lemma 4. Let the nodes be labeled as $B = \{b_1, b_2, \ldots, b_n\}$ and $L = \{\ell_1, \ell_2, \ldots, \ell_n\}$, for some $n > 1$. Nodes are placed in space so that, for some $d_{\min} > 0$ and for all $i, j \in [n]$, if $i \neq j$ it is $d(b_i, \ell_j) > d_{\min}$, or otherwise it is $d(b_i, \ell_j) = d_{\min}$, and every beeper (resp. listener) has the same distribution of influence among its outgoing (resp. incoming) links (see a deployment example in Figure 2 at the beginning of the Appendix).

Under the above definition the influence threshold is $1$ and the set of influential links is $E_1 = \{(b_i, \ell_i) | i \in [n]\}$. Recall that $W = \max_{E' \subseteq E_1} \frac{1}{|E'|} \sum_{(b,\ell) \in E'} \sum_{(b',\ell) \in E} w(b', \ell)$.

Given that all listeners have the same influence distribution among their incoming links, the whole set $E_1$ maximizes $W$, that is

$$W = \frac{1}{n} \sum_{(b,\ell) \in E_1} \sum_{(b',\ell) \in E} w(b', \ell) = \frac{1}{n} n \sum_{(b,\ell) \in E_1} w(b, \ell_1) = 1 + \sum_{(b,\ell) \in E_1 \setminus \{(b_1,\ell_1)\}} w(b, \ell_1) \,.$$

The middle equality is true because all listeners have the same distribution of influence among its incoming links. Let $T(E_1)$ be the number of time slots needed to solve One-beep Local Broadcast on $E_1$ by some protocol $\mathcal{P}$. Applying Lemma 4 to each time slot of the execution of $\mathcal{P}$, we have that for link $(b_1, \ell_1) \in E$ it is $\sum_{(b,\ell) \in E_1 \setminus \{(b_1,\ell_1)\}} w(b, \ell_1) \in O(T(E_1))$. The latter holds regardless of whether $\mathcal{P}$ uses randomization or not. Thus, replacing, we get $W \in O(T(E_1))$ with probability $1$, and the claim follows. □

## 5 DISCUSSION AND OPEN DIRECTIONS

We showed, using formal analysis, that the complexity of Neighborhood Learning in WBN networks is nearly-linearly proportional to the newly introduced characteristic – MaxAveInf. Such characteristic is global, and its range may span from a constant to the number $n$ of all agents, depending on the weighted network (even if neighborhoods are small). Our lower bound shows that executing Neighborhood Learning in parallel may be costly even in networks with small degree (but with high MaxAveInf). On the other hand, for networks with small MaxAveInf (which was also the case in most of the considered nature-driven and synthetic datasets, see Section E.3), we give a provably-efficient nearly-optimal algorithmic solution. Our theoretical bounds achieve a gap of at most $(\mathcal{M} + \log n) \log n$ between the upper bound (for messages of length $\mathcal{M}$) and lower bounds (for single-bit messages). That means that our upper bound is nearly-optimal for systems with short messages $\mathcal{M} \in O(\log n)$, yielding a gap of $O(\log^2 n)$. That is, our algorithm is well scalable. Appendix E contains experimental analysis confirming our theoretical results on diverse nature-driven and synthetic datasets. Our algorithm also outruns other competitors.

Interesting open problems include: the study of more complex nature-inspired models in the WBN framework (including binary encoding of more complex model features), more complex learning problems, the relation between the influence threshold $\sigma$ and the MaxAveInf $W$, and *provably* accurate estimation methods of the latter. We conjecture that our general lower bound could be improved for specific graph topologies, e.g., for (nearly-) uniform degree graphs.

**Reproducibility statement.** This paper contains all the details of our theoretical study, including the presentation of our algorithms, pseudocodes, and upper and lower bound proofs. Due to space constraints, the details of certain proofs have been relegated to Appendix B and D, as well as the details of algorithmic extensions to Appendix C. A full description of our extensive experimental study can be found in Appendix E, including citations to the publicly available datasets tested, as well as to an anonymous repository where we uploaded all the code and input/output data. Further details on experimental part, such as our computational platform, are included in the reproducibility section of Appendix E.

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

# APPENDIX

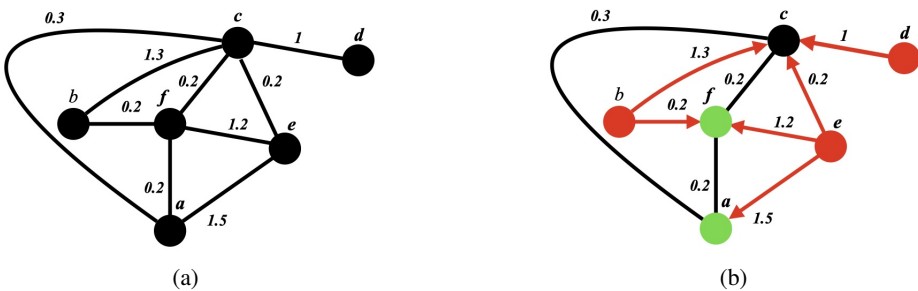

(a)                    (b)

Figure 1: Beeping communication model. Fig. (a) Example of WBN topology modeled by a bidirected weighted graph, where weights correspond to influence, see Section 2.2. The arrows of each bidirected link are omitted for clarity. Fig. (b) Example of the beep delivery concept, as defined in Sec. 2.3. The depicted arrows point from the beeping nodes to their neighbors, while the remaining arrows are omitted. Consider a time slot when nodes $b$, $d$, and $e$ beep (in red) and $a$, $f$, and $c$ listen. Node $a$ hears a single beep from $e$, hence this beep is also delivered to $e$ from $a$ (indicated in green). Node $f$ also hears a beep, caused by its two beeping neighbors: $b$, with influence $0.2$, and $e$, with influence $1.2$; hence, the beep from $e$ is delivered to $f$ (indicated in green) while the beep from $b$ is not. Node $c$ also hears a beep, caused by its three beeping neighbors: $b$, with influence $1.3$, node $e$, with influence $0.2$, and $d$, with influence $1$; as a result, no beep from any neighbor is delivered to $c$.

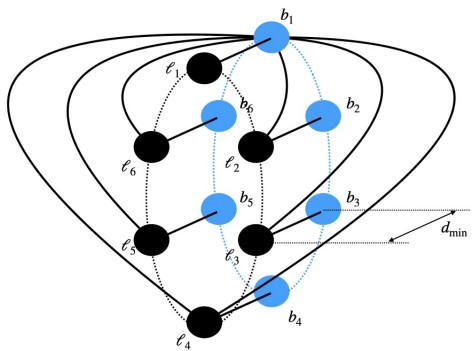

Figure 2: Example of adversarial network for $n = 6$.

## A    Limitations, Additional Motivation and Other Related Work

### A.1    Limitations of our work

From a theoretical perspective, we considered one of the simplest models of ad hoc environment with communication by beeping, WBN. We used a minimum set of model assumptions, but still we took into account weighted interactions and threshold function to separate hearing a beep from silence. Thus, our results could be transformed to more complex models and real datasets of ad hoc environments. Such transformations are, however, not straightforward and require further studies, depending on the complexity of the considered variant of the communication model. For instance, when applying our algorithm to the bee colony environment, one needs to take into account that bees communicate using complex signals called "waggle dance"; it could be encoded into a binary sequence, see Dong et al. (2023), however a future study should take into account that not every binary sequence represents a valid waggle dance (i.e., recognized by bees).

### A.2    Applications of WBN to Better Understanding Collective Intelligence of Bacteria

Our WBN model can be seen as an abstraction of chemical communication mechanisms used by bacteria. Specifically, in *quorum sensing* Moreno-Gámez et al. (2023) bacteria release/sense some *autoinducers* ($\equiv$ beeps) to the intercellular space. Upon reaching a threshold concentration ($\equiv$ influence threshold), autoinducers trigger cascades of *signal transduction* which regulate processes such as biofilm formation, virulence, competence and sporulation Miller & Bassler (2001); Fuqua et al. (1994); Waters & Bassler (2005). Cooperation among bacteria as a multi-cellular (multiagent) system has been here for billions of years. A connection between WBN and bacteria colonies may lead to a better understanding of their collective intelligence, as well as interactions with (and collaborative influence to) surrounding cells of other organisms. For instance, next step could be to extend the WBN model to different types of beeps in order to model even more complex signal transduction processes (see Galperin (2004)).

### A.3    Other Related Work

Solving NL is crucial for enabling effective cooperation and scalability in MAS, particularly in complex and large-scale scenarios where agents only have partial observations of the environment. By focusing on local interactions, agents can achieve coordination and collective goals without needing a full, centralized view, which is often infeasible due to computational limits or communication constraintsOroojlooy & Hajinezhad (2023). NL allows agents to maintain neighborhood cognitive consistency, leading to improved system-level cooperation by ensuring neighboring agents' internal representations about their local environment are aligned Mao et al. (2020). This decentralized approach is particularly effective in Multi-Agent Reinforcement Learning (MARL) tasks, where agents can leverage local information sharing to mitigate issues like non-stationarity and partial observability.

In Neighborhood Cognitive Consistency for instance, agents maintain coordinated policies based on joint observations within their local group, which is sufficient for effective system-level cooperation in large-scale settings like packet routing and Google Football player control Mao et al. (2020). This local focus provides a scalable alternative to centralized training approaches, where a single node handles all agents and suffers from performance degradation as the number of agents increases Malucelli (2024). Furthermore, in tasks like Swarm Control and other cooperative settings, NL can be used to define intrinsic rewards based on the prediction error of aggregated neighboring features – a strategy called Neighborhood-curiosity-based Exploration – which motivates agents to explore new states and discover effective cooperative partnerships Yang et al. (2024). Additionally, training on alternating, large subsets of agents (neighborhoods), a technique known as Large Neighborhood Search, significantly improves training efficiency and speeds up convergence in complex cooperative MARL tasks like the StarCraft Multi-Agent Challenge Chen et al. (2024b). Another example is decentralized training, where NL can be used to average or aggregate model parameters or stochastic gradients among neighboring nodes in the communication network. Instead of communicating with a central server, each device computes its local update and then uses the NL subroutine to exchange and combine this information only with its one-hop neighbors. The repeated application of this neighborhood averaging step helps the locally trained models across all devices to converge toward a consensus (a shared "global" model) in a distributed manner, similar to a Distributed Parallel Stochastic Gradient Descent.

A related problem of local broadcast has been recently studied in simple beeping networks (without weights and thresholds), although there is still a polynomial (in terms of the maximum node degree) gap between the best known upper and lower bounds' formulas, see Beauquier et al. (2018); Davies (2023) resp.

From a technical perspective, the closest related work is on link scheduling in wireless networks. The similarities include: ad hoc network structure and potential impact of one node communication to the other. However, the problem of link scheduling is focused on realization of specific known links (typically, *one* link per node), unlike in Neighborhood Learning where the information needs to be propagated via *all* (unknown) links with sufficiently large influence. Also, the wireless communication is focused on avoiding interference, while in WBN we want to "strengthen" the information beeped via links of large influence.

The closest wireless Link Scheduling under interference related works Halldórsson & Wattenhofer (2009); Fanghänel et al. (2009; 2011); Kesselheim & Vöcking (2010); Kesselheim (2012); Ásgeirsson et al. (2017); Halldórsson & Wattenhofer (2019) correspond to the Signal-to-Interference-and-Noise-Ration (SINR) interference model. They use a few different characteristics, called affectance, and their impact to link scheduling performance, see e.g., Halldórsson & Wattenhofer (2009). The *generalized affectance* model was introduced and used only in the context of one-hop communication, more specifically, to Link Scheduling by Kesselheim et al. Kesselheim (2012); Kesselheim & Vöcking (2010). In Kowalski et al. (2014; 2023), the one-hop affectance characteristic was generalized, called the *maximum average tree-layer affectance*, to be applicable to multi-hop communication tasks such as broadcast, together with another characteristic, called the *maximum path affectance*. In a more recent summary Halldórsson & Wattenhofer (2019) of work in Wireless Networks over the last decade theoretical studies of Link Scheduling are overviewed.

# B    FULL PROOF OF LEMMA 1 FROM SECTION 3.1

*Proof.* For a given local broadcast protocol, let $X_x(t)$ be a random variable indicating whether node $x \in V$ beeps at time $t$, and let $X_{x,y}(t)$ be a random variable indicating whether a beep from $x$ was delivered to $y$ at time $t$. In terms of this notation, the claim to prove is

$$\mathbf{E}\left(\sum_{(x,y)\in E'_\sigma} X_{x,y}(t)\right) \geq |E'_\sigma|(1-p)p/4.$$

By linearity of expectation, we have that

$$\mathbf{E}\left(\sum_{(x,y)\in E'_\sigma} X_{x,y}(t)\right) = \sum_{(x,y)\in E'_\sigma} \mathbf{E}\left(X_{x,y}(t)\right). \tag{3}$$

Let $M \subseteq E'_\sigma$ be the set $M = \{(x,y) \in E'_\sigma | \sum_{z \in V(t)} w(z,y) \leq 2W\}$. Then we can re-write Equation 3 as

$$\mathbf{E}\left(\sum_{(x,y) \in E'_\sigma} X_{x,y}(t)\right) = \sum_{(x,y) \in M} \mathbf{E}\left(X_{x,y}(t)\right) + \sum_{(x,y) \in E'_\sigma \setminus M} \mathbf{E}\left(X_{x,y}(t)\right)$$

$$\geq \sum_{(x,y) \in M} \mathbf{E}\left(X_{x,y}(t)\right). \tag{4}$$

On the other hand recall that, for any $(x,y) \in E$, the conditions to deliver a beep from node $x$ to node $y$ at time $t$ are $X_y(t) = 0$, $w(x,y)X_x(t) \geq \sigma$, and $\sum_{z \in V(t):z \neq x} w(z,y)X_z(t) < 1$. Then, we have that

$$\mathbf{E}\left(X_{x,y}(t)\right) = Pr(X_{x,y}(t) = 1)$$
$$= Pr(X_x(t) = 1) \cdot Pr(X_y(t) = 0) \cdot Pr(X_{x,y}(t) = 1 | X_x(t) = 1 \wedge X_y(t) = 0)$$
$$= p(1-p)Pr(X_{x,y}(t) = 1 | X_x(t) = 1 \wedge X_y(t) = 0).$$

Replacing the latter in Equation 4, we have that

$$\mathbf{E}\left(\sum_{(x,y) \in E'_\sigma} X_{x,y}(t)\right) \geq p(1-p) \sum_{(x,y) \in M} Pr(X_{x,y}(t) = 1 | X_x(t) = 1 \wedge X_y(t) = 0)$$

$$= p(1-p) \sum_{(x,y) \in M} \left(1 - Pr(X_{x,y}(t) = 0 | X_x(t) = 1 \wedge X_y(t) = 0)\right). \tag{5}$$

From the conditions to deliver a beep we have that, for any $(x,y) \in M$, it is

$$Pr\left(X_{x,y}(t) = 0 | X_x(t) = 1 \wedge X_y(t) = 0\right) = Pr\left(\sum_{z \in V(t):z \neq x} w(z,y)X_z(t) \geq 1\right).$$

And by Markov inequality [10],

$$Pr\left(X_{x,y}(t) = 0 | X_x(t) = 1 \wedge X_y(t) = 0\right) \leq \mathbf{E}\left(\sum_{z \in V(t):z \neq x} w(z,y)X_z(t)\right)$$

$$= \sum_{z \in V(t):z \neq x} w(z,y)p \leq p \sum_{z \in V(t)} w(z,y).$$

Replacing in the latter that $\sum_{z \in V(t)} w(z,y) \leq 2W$ (by definition of $M$) and that $p \leq 1/(4W)$ (by the condition of the lemma), it is

$$Pr\left(X_{x,y}(t) = 0 | X_x(t) = 1 \wedge X_y(t) = 0\right) \leq 1/2.$$

Replacing the latter in Equation 5, we have that

$$\mathbf{E}\left(\sum_{(x,y) \in E'_\sigma} X_{x,y}(t)\right) \geq |M|p(1-p)/2. \tag{6}$$

Finally, we lower bound $|M|$ as follows. Recall that

$$W = \max_{E' \subseteq E_\sigma} \frac{1}{|E'|} \sum_{(x,y) \in E'} \sum_{z \in V(t)} w(z,y),$$

and that

$$M = \{(x,y) \in E'_\sigma | \sum_{z \in V(t)} w(z,y) \leq 2W\}.$$

Then, by Pigeonhole principle, it is $|M| \geq |E'_\sigma|/2$. Replacing in Equation 6, the claim of the lemma follows. $\qquad \square$

---

[10]Notice that Markov inequality does not require independence of the random variables (see Theorem 3.1 in Mitzenmacher & Upfal (2005)).

# C  DETAILS ON EXTENSIONS OF THE MAIN ALGORITHM, FROM SECTION 3.3

**Arbitrary (unknown) $W$ and only polynomial upper bound on $n$.**  First, observe that in the algorithm we only need a polynomial upper bound on $n$, as $n$ occurs under logarithm in the formula on MaxAveInf in Theorem 2.

The impact of an estimation of the actual parameter $W$ could be, hypothetically, more significant, as the number of learning rounds depends linearly on $W$ (see Thm. 2 and later Thm. 3). Our algorithm could be transformed into one without the need of knowing $W$ in advance, by using doubling estimates of $W$ (i.e., we keep running our algorithm for parameters $W$ being subsequent powers of 2) until no delivery. The main property that makes it feasible is described in Lemma 3 – receiving agents can recognize whether the current communication attempt comes from one agent (which is desired) or more than one (in which case, the received sequence of beeps can be ignored, hence – no delivery). Additionally, acknowledgments of successful transmissions have to be implemented: by repeating the same algorithm, but this time with messages being acknowledgments of previously received messages. Note that this method increases the number of steps only by a constant factor.

For the sake of our experiments in Appendix E, we designed a stochastic estimation of $W$, which is quite accurate on the tested inputs, without significant affectance on algorithm performance.

**De-synchronized systems.**  Due to a relatively simple structure and various checking mechanisms in our algorithm, it can be extended to de-synchronized settings (with clock shifts) by using the known synchronizing methods for beeping networks, see e.g., the recent work De Marco & Kowalski (2025). These methods require only a constant overhead, and transform algorithms with structures similar to ours (i.e., main loop iterating a preamble code and message) from fully synchronized into de-synchronized solutions.

**Additional comments on asynchrony.**  There are natural multiagent environments where the entities have an initial phase of synchronization followed by synchronous interactions. For instance, bacteria communicate with each other through a process called quorum sensing, using chemical signals to coordinate group behaviors. This communication allows them to sense population density and synchronize activities like biofilm formation, virulence, and bioluminescence.

We also note that full asynchrony (i.e., when a single beep cycle of one agent could correspond to several beeps of other agent(s)) is a challenging question, left to future work, because even at starting point it requires defining the feedback based not just on the current set of transmitters, but actually on transmitting sequences (of potentially different lengths) of many simultaneously active agents. From a practical perspective, since we account for every single communication bit in the beeping model, clock drifts and asynchrony would imply that entities try to communicate using different frequencies, which does not typically happen in natural or applied systems, and if it happens, it typically implies no successful communication means.

More precisely, our model is a low-level information-theoretical model. Unlike a vast majority approaches, e.g., federated or distributed learning, which operate at high layers of the protocol stack (and thus have to deal with asynchrony), our approach models low-layer algorithms. By analogy to communication protocols' stack, our algorithmic approach combines coding (physical layer) with overcoming negative influence (link layer), and our goal was to characterize efficiency of learning at such low (information) level. While clock shifts are natural phenomena for low-level protocols, asynchrony (as understood in classic distributed computing/learning) is not. The reason is that asynchrony at low level would mean in fact different frequency, which in case of both artificial wireless networks and biological networks is typically considered non-compatible. For instance, while codes could be applied to higher-level asynchronous algorithms in distributed computing, the codes themselves are typically designed and analyzed as synchronous objects (i.e., with fixed position indexing by subsequent integers).

**Dynamic weights and noise.**  In our model, physical conditions, such as noise, are embedded in link weights. The question of fluctuating physical conditions then boils down to how dynamic weights impact the complexity of NL. Under arbitrary dynamicity, NL is not computable in a MAS such as a WBN: if neighborhoods may change in every round, the computation must have taken at most one round. This observation yields that NL could be defined for restricted dynamic settings,

for instance, if weights change only every so many rounds, or changes are driven by a stochastic process with bounded deviation. Our algorithms can be applied to such settings as well, as long as weights are locally static (or nearly static, in the sense that at any time point, their MaxAveInf does not exceed a given upper bound) during the execution of the algorithm. This is because Lemma 1 still holds for any single round of such dynamic model, while the way it is used in the final proofs of Theorem 1 and Theorem 2 is independent on specific values of weights but depends only the upper bound on MaxAveInf. More dynamic scenarios, e.g., when MaxAveInf of current weights may oscillate, are left as a challenging open problem. Note that our lower bound (for static weights) automatically extends to dynamic scenarios, and thus gives a point of reference to the future study of more dynamic scenarios.

# D  FULL PROOF OF LEMMA 4 FROM SECTION 4

*Proof.* To prove the claim we split the set of links $\mathcal{E}$ into *distant* and *close* with respect to listener $\ell'$. For the set $\mathcal{E}_{close}$ of close links, we upper bound the influence of their beepers on $\ell'$ simply by $|\mathcal{E}_{close}|$ (since each link has influence at most 1), and we bound $|\mathcal{E}_{close}|$ by a constant using a geometric argument and the fact that a beep is delivered through every link in $\mathcal{E}_{close}$. For the set $\mathcal{E}_{dist}$ of distant links, we bound the influence of their beepers on $\ell'$, by their influence on a listener at shortest distance from $\ell'$. We bound the latter by a constant using a geometric argument and the fact that a beep is delivered through every link in $\mathcal{E}$. The details follow.

Let $2r$ be the shortest distance from any listener in $\mathcal{E}$ to listener $\ell'$. Let $\mathcal{E}_{close} = \{(b,\ell) \in \mathcal{E} | d(b,\ell') \leq r\}$, that is the set of links $(b,\ell) \in \mathcal{E}$ whose beeper $b$ is within distance $r$ from listener $\ell'$.

We bound first the influence of close beepers on $\ell'$ as

$$\sum_{(b,\ell) \in \mathcal{E}_{close}} w(b,\ell') = \sum_{(b,\ell) \in \mathcal{E}_{close}} \frac{d_{\min}}{d(b,\ell')} \leq |\mathcal{E}_{close}| . \tag{7}$$

We now bound $|\mathcal{E}_{close}|$ as follows. Fix a link $(b'',\ell'') \in \mathcal{E}_{close}$. The aggregated influence of the other links in $\mathcal{E}_{close}$ on listener $\ell''$ is

$$\sum_{(b,\ell) \in \mathcal{E}_{close}\setminus\{(b'',\ell'')\}} w(b,\ell'') = \sum_{(b,\ell) \in \mathcal{E}_{close}\setminus\{(b'',\ell'')\}} \frac{d_{\min}}{d(b,\ell'')} . \tag{8}$$

The equality in the latter is due to the graph being complete bipartite. The boundary case of the following geometric argument is illustrated in Figure 3a.

Given that $b''$ is within distance at most $r$ of $\ell'$ (because $(b'',\ell'') \in \mathcal{E}_{close}$), and $\ell''$ is within distance at least $2r$ from $\ell'$ (because $2r$ is the shortest distance from any listener in $\mathcal{E}$ to $\ell'$), the length of $(b'',\ell'')$ is at least $r$ (i.e. $r \leq d(b'',\ell'')$). Also, given that $b$ and $b''$ are both within distance $r$ of $\ell'$ (because $(b,\ell) \in \mathcal{E}_{close}$ and $(b'',\ell'') \in \mathcal{E}_{close}$), we have that $b$ is within distance $2r$ of $b''$ (i.e. $d(b,b'') \leq 2r$). Combining and using the triangle inequality, we have that

$$d(b,\ell'') \leq d(b,b'') + d(b'',\ell'') \leq 2r + d(b'',\ell'') \leq 3d(b'',\ell'') .$$

Replacing in Equation 8,

$$\sum_{(b,\ell) \in \mathcal{E}_{close}\setminus\{(b'',\ell'')\}} w(b,\ell'') \geq \frac{|\mathcal{E}_{close}| - 1}{3} \frac{d_{\min}}{d(b'',\ell'')} = \frac{|\mathcal{E}_{close}| - 1}{3} w(b'',\ell'') . \tag{9}$$

On the other hand, given that a beep is delivered through each link in $\mathcal{E}$, in particular a beep is delivered through link $(b'',\ell'')$. Therefore, using the conditions for successful delivery we know that

$$\sum_{(b,\ell) \in \mathcal{E}_{close}\setminus\{(b'',\ell'')\}} w(b,\ell'') < w(b'',\ell'') .$$

Replacing the latter in Equation 9, it is $(|\mathcal{E}_{close}| - 1)w(b'',\ell'')/3 < w(b'',\ell'')$, that is, $|\mathcal{E}_{close}| < 4$. Replacing in Equation 7, we get

$$\sum_{(b,\ell) \in \mathcal{E}_{close}} w(b,\ell') < 4. \tag{10}$$

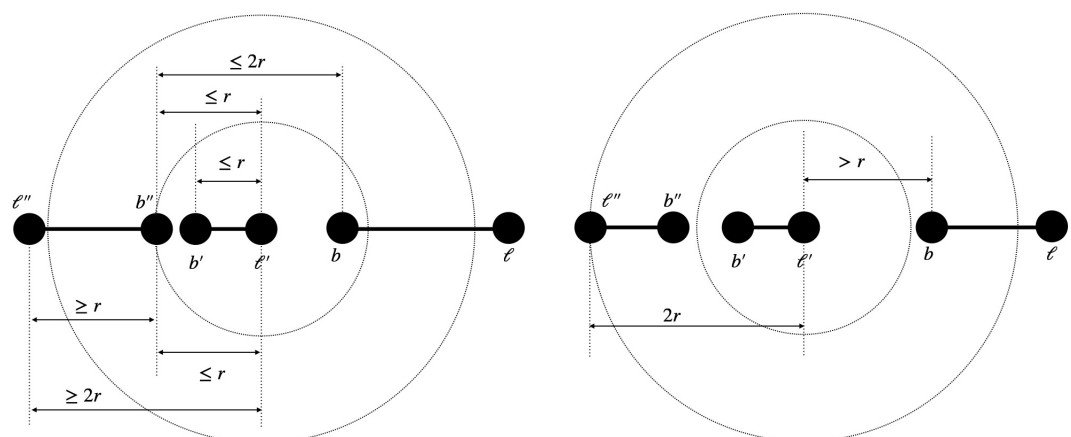

(a) Distance bound $d(b, \ell'') \leq 3d(b'', \ell'')$, for some link $(b'', \ell'') \in \mathcal{E}_{close}$, which yields $\sum_{(b,\ell) \in \mathcal{E}_{close}} w(b, \ell') < 4$ using that a beep is delivered through link $(b'', \ell'')$.

(b) Distance bound $d(b, \ell'') \leq 3d(b, \ell')$, for some link $(b'', \ell'') \in \mathcal{E}$ whose listener is at shortest distance $2r$ of $\ell'$, which yields $\sum_{(b,\ell) \in \mathcal{E}_{dist}} w(b, \ell') < 6$ using that a beep is delivered through link $(b'', \ell'')$.

Figure 3: Illustration of distance bounds.

We now bound the influence of beepers in $\mathcal{E}_{dist} = \mathcal{E} \setminus \mathcal{E}_{close}$ on listener $\ell'$. We have that

$$\sum_{(b,\ell) \in \mathcal{E}_{dist}} w(b, \ell') = \sum_{(b,\ell) \in \mathcal{E}_{dist}} \frac{d_{\min}}{d(b, \ell')}. \tag{11}$$

The equality in the latter is due to the graph being complete bipartite. The boundary case of the following geometric argument is illustrated in Figure 3b.

Let $(b'', \ell'') \in \mathcal{E}$ be a link such that $\ell''$ is a listener at shortest distance $2r$ from listener $\ell'$. By triangle inequality, for any $(b, \ell) \in \mathcal{E}_{dist}$, it is $d(b, \ell'') \leq d(b, \ell') + d(\ell', \ell'') = d(b, \ell') + 2r$. Given that, for any $(b, \ell) \in \mathcal{E}_{dist}$, it is $r \leq d(b, \ell')$, we have that $d(b, \ell'') \leq 3d(b, \ell')$. Replacing $d(b, \ell') \geq d(b, \ell'')/3$ in Equation 11, we get

$$\sum_{(b,\ell) \in \mathcal{E}_{dist}} w(b, \ell') \quad \leq \quad \sum_{(b,\ell) \in \mathcal{E}_{dist}} \frac{3d_{\min}}{d(b, \ell'')} \quad = \quad 3 \sum_{(b,\ell) \in \mathcal{E}_{dist}} w(b, \ell'')$$

$$\leq \quad 3 \left( w(b'', \ell'') + \sum_{(b,\ell) \in \mathcal{E}_{dist} \setminus \{(b'', \ell'')\}} w(b, \ell'') \right) .$$

Given that a beep is delivered through each link in $\mathcal{E}$, in particular it is delivered through $(b'', \ell'')$. Hence, by the condition to have a beep delivered through the link $(b'', \ell'')$, it is $\sum_{(b,\ell) \in \mathcal{E}_{dist} \setminus \{(b'', \ell'')\}} w(b, \ell'') < 1$. On the other hand, it is $w(b'', \ell'') \leq 1$ because for the input topology used in this proof all $w(\cdot, \cdot)$ are at most 1. Replacing in the latter we get

$$\sum_{(b,\ell) \in \mathcal{E}_{dist}} w(b, \ell') < 6 . \tag{12}$$

Combining the bounds in Eqs. 10 and 12 the claim follows. $\qquad \square$

## E  EXPERIMENTAL EVALUATION

To evaluate experimentally the performance of our algorithm we have considered real-world social networks (extracted from nature) on top of which nano-devices could be deployed, and to complement those we also considered ad-hoc infrastructures, which could be deployed for instance for 3D scanning and learning simultaneously.

All input datasets used in our simulations are either publicly available in Rossi & Ahmed (2015), or in our anonymous private repository in https://anonymous.4open.science/r/OBLB-A8F8/, where we also include all code and output files. The latter repository will be de-anonymized upon publication of this work.

For natural environments, we studied performance for four datasets: a population of birds Firth & Sheldon (2015), a colony of ants Mersch et al. (2013), a population of dolphins Gazda et al. (2015), and a mouse gene regulatory network derived from analyzing gene expression profiles Bansal et al. (2007). For the latter we extracted various subgraphs (as detailed in Table 1) to process the original dataset ($|E| \approx 14.5M$). These inputs are representative of different network sizes (in nodes and links) and weights.

Link weights in those datasets correspond to the probability or frequency of interaction for each pair of individuals. Conceptually, these weights match our definition of influence, because an individual $a$ that meets more frequently another individual $b$ "influences" more the behavior of $b$. In the rest of the section we use *influence* and *weight* indistinctively.

These natural datasets observe an exponential distribution of weights, biased towards small probabilities/frequencies (as in many natural interactions). Given that the threshold for influential links is $\sigma \geq 1$, and that the distributions are biased towards small values, we normalized all weights to a range $[0, 10]$.

The real-world datasets were downloaded from the public repository in Rossi & Ahmed (2015) at

- Birds: https://networkrepository.com/aves-wildbird-network.php
- Ants: https://networkrepository.com/insecta-ant-colony6-day04.php
- Dolphins: https://networkrepository.com/mammalia-dolphin-florida-forage.php.
- Genes: https://networkrepository.com/bio-mouse-gene.php.

| dataset | $|V|$ | $|E|$ | original weights | normalized to range |
|---|---|---|---|---|
| birds: Firth & Sheldon (2015) | 202 | 11900 | half-weight index in $[0, 1]$ | $[0, 10]$ |
| ants: Mersch et al. (2013) | 164 | 10300 | interaction count in $[1, 97]$ | $[0, 10]$ |
| dolphins: Gazda et al. (2015) | 190 | 1100 | interaction count in $[1, 7]$ | $[0, 10]$ |
| genes: Bansal et al. (2007) | 1000 | 90881 | probabilistic interaction | $[0, 10]$ |
| | 2000 | 258909 | | |
| | 3000 | 432239 | | |
| | 4000 | 563033 | | |
| | 5000 | 675129 | | |
| synthetic scale-free network | 1000 | 3994 | exp. dist. in $[0, 1]$ | $[0, 10]$ |
| | 2000 | 7994 | | |
| | 3000 | 11994 | | |
| | 4000 | 15994 | | |
| | 5000 | 19994 | | |
| synthetic 3D grid | 216 | 1078 | exp. dist. in $[0, 1]$ | $[0, 2^i]$, $i = [0, 7]$ |

Table 1: Networks evaluated. Half-weight index = probability that two individuals are observed together given that one has been seen. Interaction count = number of times in spatial proximity within the time span of data collection.

To complement our real-world scenarios, we also evaluated two ad-hoc networks: a scale-free network and a 3D-topology. To generate the scale-free network we used the preferential attachment model of Barabási & Albert (1999), obtaining various input graphs with $|V| = 1000, 2000, 3000, 4000, 5000$ and $|E| = 3994, 7994, 11994, 15994, 19994$ respectively. For the second we created a 3D-grid of $6 \times 6 \times 6$ vertices placing a node in each intersection, which yields $|V| = 216$ and $|E| = 1078$. Weights were defined following an exponential distribution similar to the nature datasets. Specifically, for in the 3D grid each weight was drawn uniformly at random

| dataset | $\sigma$ | $w_{limit}$ |
|---|---|---|
| birds: Firth & Sheldon (2015) | $5, 4.8, 4.6, \ldots, 1$ | $5, 6, \ldots, 10$ |
| ants: Mersch et al. (2013) | $3, 2.8, 2.6, \ldots, 1$ | $4, 5, \ldots, 10$ |
| dolphins: Gazda et al. (2015) | $5, 4.8, 4.6, \ldots, 1$ | $6, 7, \ldots, 10$ |
| genes: Bansal et al. (2007) | $5, 4, \ldots, 1$ | $10$ |
| synthetic scale-free network | $1.5, 1.3, 1.1$ | $10$ |
| synthetic 3D grid | $1.5, 1.3, 1.1$ | none |

Table 2: Simulation parameters.

a number in $(i, i + 0.1]$ with probability $1/2^i$, for each $i = 0, 0.1, \ldots, 0.9$; and for the scale-free network we used a random exponential distribution with rate parameter 5.

The main characteristics of the networks evaluated and the parameter values chosen are listed in Tables 1 and 2 respectively.

### E.1 How to compute $W$

For the datasets evaluated, MaxAveInf $W$ is unknown. Computing $W$ exactly would be prohibitively time consuming because it requires to maximize some average over *all* subsets of influential links. Our algorithm can be extended to handle the issue implementing an exponential search of $W$ (doubling estimates), as explained in Appendix C. Nevertheless, to focus on the algorithmic dependency on $W$ and $E_\sigma$, we developed experimentally an estimation method, which is an interesting problem on its own. Namely, for each input dataset, we estimated $W$ randomly sampling the set of influential links in size and content. To improve accuracy, we repeated this estimate 200 times, keeping the largest $W$ obtained. An appropriate number of times to repeat the calculation, so that the chosen value (200) yields an accurate estimation, was experimentally determined trying increasing numbers until convergence is reached. As seen in Figure 4 for some of our inputs, the estimate converges rapidly even with a much smaller number of repetitions [11].

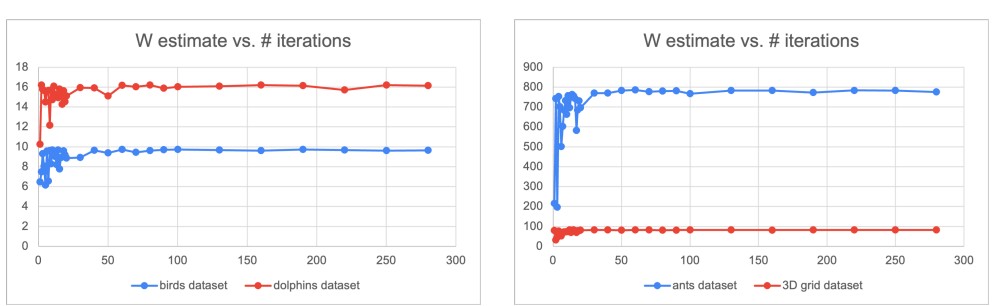

Figure 4: MaxAveInf estimate versus number of repetitions of the estimation for all datasets studied.

Another challenge is how to obtain various inputs with different $W$ values (based on the same dataset) so that we can evaluate the dependency of time on $W$. For the genes dataset, we addressed this challenge using decreasing values of $\sigma$ from an initial larger value ($\sigma_{limit} \leq 10$) down to 1. A changing $\sigma$ results in a changing $|E_\sigma|$ (the number of influential links), which possibly yields a changing $W$ (see the definition of $W$ in Section 2.4). For the birds, ants, and dolphins datasets, we additionally used subsets of links of the input graph, adding only the links with weight up to some limit $w_{limit}$, from some $w_{limit} \geq \sigma_{limit}$ up to the maximum weight 10. The latter handles the case where the distribution of weights is such that the end points of links with large weight have also large weighted degree (e.g. ants dataset). Then, starting with a large $\sigma_{limit}$ (hence, small $|E_\sigma|$) would not change significantly $W$, even for larger $|E_\sigma|$. For the synthetic datasets (i.e., 3D-grid and scale-free networks) on the other hand, we induced the increasing values of $W$ adding a multiplicative factor of $1, 2, 4, \ldots, f$ to the weights, where $f = 128$ for the 3D grid and $f = 8$ for the scale-free networks,

---

[11]Notice that our goal in this work is not to provide a provable, and/or decentralized, estimator for the MaxAveInf - which is an interesting open question on its own.

for $\sigma = 1.5, 1.3$, and $1.1$. All those ranges were determined experimentally to attain enough variety of values of $W$.

For the values of $W$ obtained, we evaluated the performance of our *One-beep Local Broadcast (OBLB)* algorithm as defined in Section 3. Evaluating OBLB is enough for the purpose of evaluating the dependency of our Neighborhood Learning algorithm on $W$, because the overhead of Neighborhood Learning with respect to OBLB is only logarithmic on $n$ (for messages of logarithmic size).

## E.2 BASELINE ALGORITHMS

To the best of our knowledge, there is no other One-beep Local Broadcast (or Neighborhood Learning) algorithm that runs in the beeping model. Thus, for experimental comparison with other solutions, we considered an algorithm in which nodes start execution with random delays in $[n]$, and then beep with a period equal to their own ID, which we call the *Periodic* algorithm. We tested both algorithms on the same inputs and parameter combinations. The results are shown in Figures 5, 6, and 7.

One may ask why not using a simple round-robin schedule which would solve the problem in time $n$. The shortcoming of such approach of course is that for a massive network the running time would be very slow, which is intuitively unnecessary for a local computation problem such as Neighborhood Learning. Moreover, a round-robin algorithm would require the nodes to know the global time and $n$. Nevertheless, the question that follows is how OBLB's running time scales with $n$ in practice. To answer that question we evaluated OBLB experimentally on a synthetic 3D-grid as defined above but with growing sides $6, 7, \ldots, 15$, thus $n = 6^3, 7^3, \ldots, 15^3$. Weights were defined with an exponential distribution on $[0, 10]$ biased to small values as before. To maintain $W$ and $|E_\sigma|$ "stable" under a growing $n$ we set $\sigma = 1$ initially, increasing it by $0.5$ each time that we increased $n$. The result was that during the whole simulation $W$ and $|E_\sigma|$ oscillated but stayed within the ranges $[10, 22]$ and $[402, 692]$ respectively, with $n$ growing monotonically from 216 to 3375. The results are shown in Figure 8.

To measure running time we counted rounds of communication until a beep was delivered in all influential links. All executions were repeated 20 times for each parameter combination. The seed for all random samples was 31277847.

## E.3 DISCUSSION OF RESULTS

The results of our experiments show that performance of our OBLB in practice is similar or better than the theoretical bound, whereas the Periodic algorithm is worse than the same bound. We also show that OBLB running time is independent of $n$. We have illustrated our results in box-and-whisker charts to show the statistical behavior [12] (recall that each execution was repeated 20 times). Notice that, in comparison with network size, our datasets comprise a variety of $W$ values, from comparable with $|V|$ (ants) to very small (all others for original weights).

The charts for the birds dataset in Figure 5a show on the left that, as $W$ grows, the performance of OBLB gets much better than $4W \log |E_\sigma|$ (with the upper quartile of the former smaller than the lower quartile of the latter), whereas on the right we can see that the Periodic algorithm performs similar or worse than the same function. Similar observations apply to the dolphins dataset in Figure 5b, where in fact the difference in favor of OBLB is larger. For the ants and mouse genes datasets, we can see in Figures 6a and 6b respectively much better performance of OBLB (left) than the theoretical $4W \log |E_\sigma|$ time, whereas the Periodic algorithm (right) took more than 30000 rounds for most parameter combinations.

The performance on synthesized topologies, scale-free networks and 3D grids, can be seen in Figure 7, where the same observations apply: the running time of OBLB is almost always below $4W \log |E_\sigma|$ (with a larger difference as $W$ grows), while the running time of the Periodic algorithm is above (many times above 30000 rounds, when we stopped the simulation).

---

[12] Although it is not random, the theoretical bound of $4W \log |E_\sigma|$ sometimes varies for fixed $W$ when $|E_\sigma|$ changes.

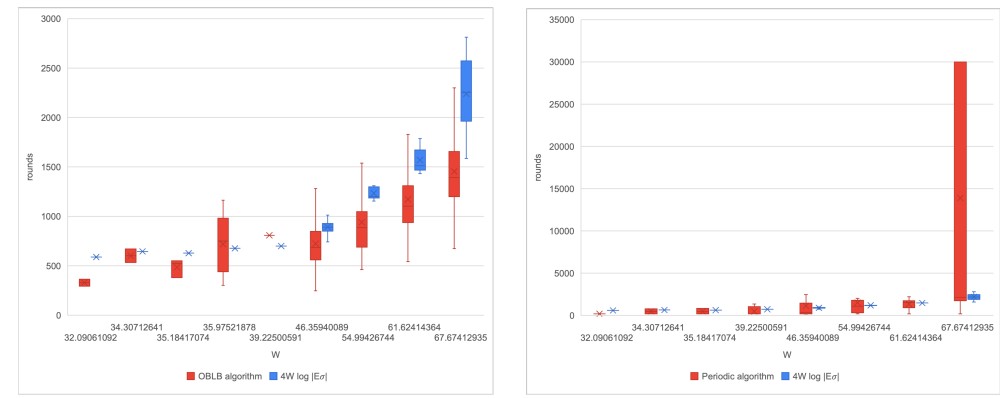

(a) Birds population dataset.

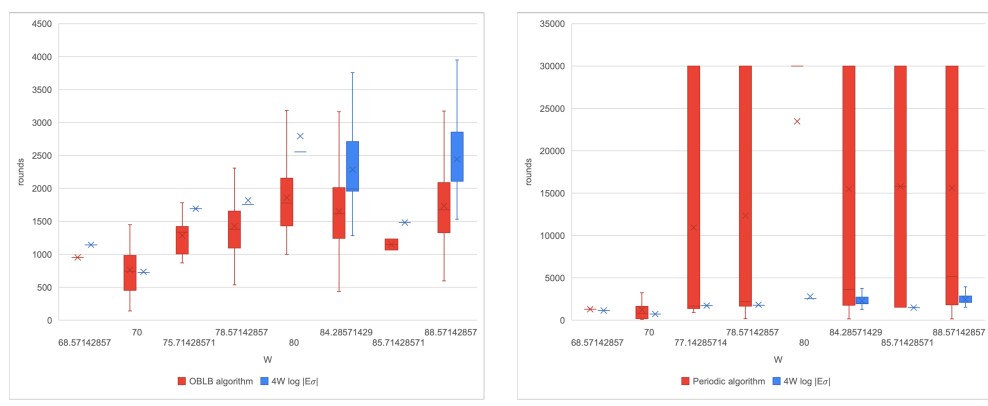

(b) Dolphins population dataset.

Figure 5: Experimental results on nature-extracted datasets in comparison with OBLB theoretical running time of $4W \log |E_\sigma|$. Left: OBLB Algorithm, right: Periodic algorithm. Statistical behavior over 20 executions for each parameter combination. All executions capped at 30000 rounds. That is, datapoints shown at 30000 rounds in fact indicate a running time of $\geq 30000$ rounds.

For the evaluation of the dependency of OBLB on $n$, we can see in Figure 8 that indeed it is independent (as the theoretical analysis showed). That is, for $n$ growing monotonically from 216 to 3375 the average running time of OBLB oscillated within a range of $[330, 580]$. That oscillation is due to the oscillation of $W$ and $|E_\sigma|$, which can be seen in the plot of the theoretical $4W \log |E_\sigma|$ included for comparison in the same chart.

E.4 REPRODUCIBILITY OF EXPERIMENTS

The nature-extracted datasets are publicly available in the repositories cited above. All synthesized datasets are included in an anonymous private repository at https://anonymous.4open.science/r/OBLB-A8F8/ and will be made publicly available upon publication of the paper.

The simulators of OBLB and Periodic algorithms were coded in Java language, compiled in the Java SE Runtime Environment (build 1.8.0_121-b13), and executed in a Java HotSpot 64-Bit Server VM (build 25.121-b13, mixed mode). All code is included in an anonymous private repository at https://anonymous.4open.science/r/OBLB-A8F8/ and will be made publicly available upon publication of the paper.

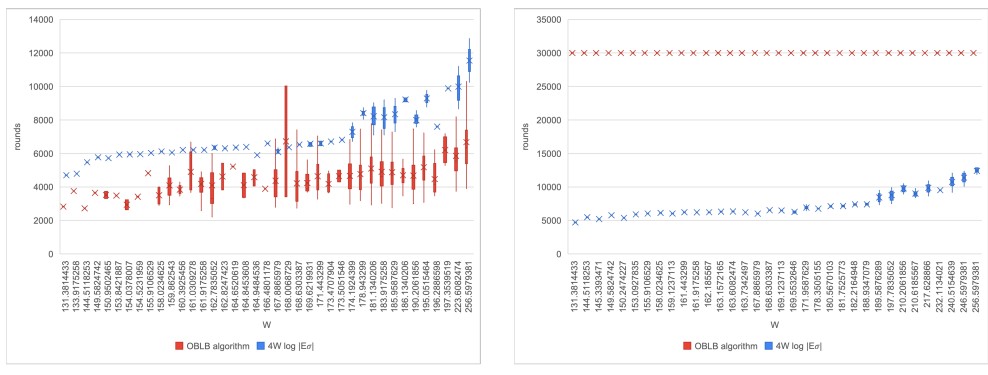

(a) Ant colony dataset.

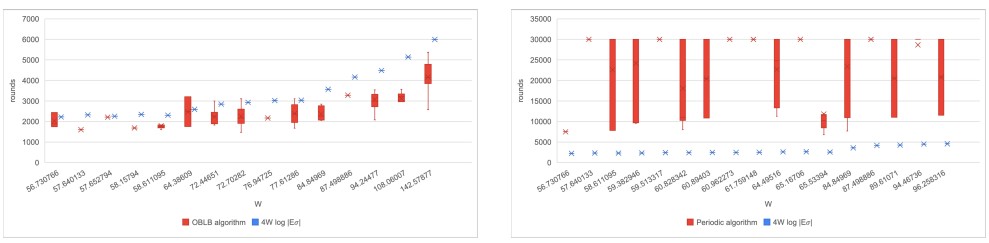

(b) Mouse genes dataset ($|V| = 1000, |E| = 90881$ shown, larger graphs produced similar results.).

Figure 6: Experimental results on nature-extracted datasets in comparison with OBLB theoretical running time of $4W \log |E_\sigma|$. Left: OBLB Algorithm, right: Periodic algorithm. Statistical behavior over 20 executions for each parameter combination. All executions capped at 30000 rounds. That is, datapoints shown at 30000 rounds in fact indicate a running time of $\geq 30000$ rounds.

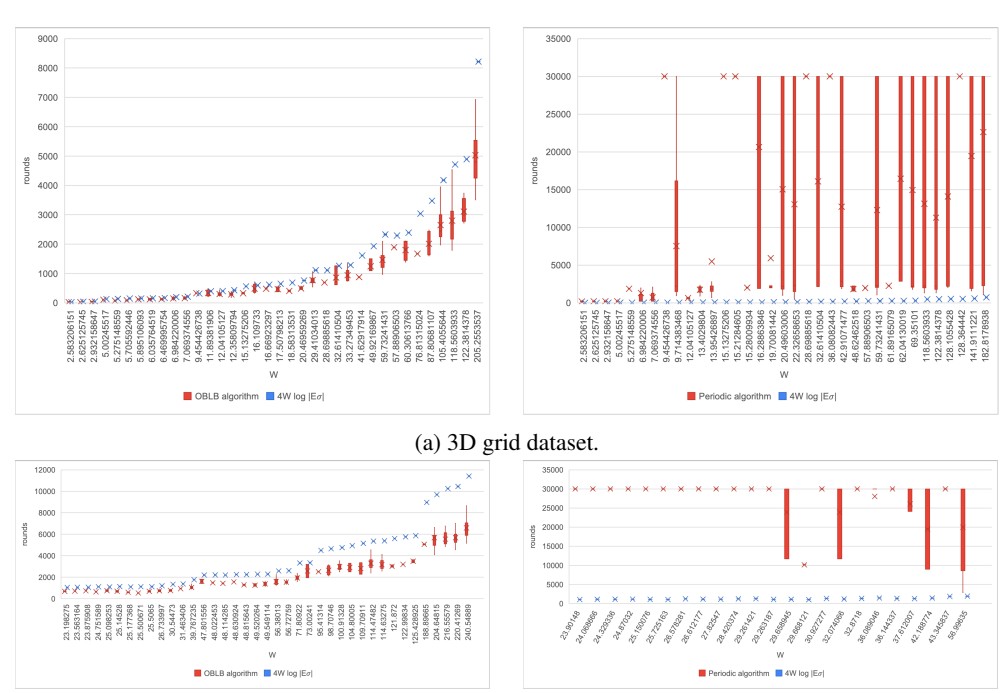

(a) 3D grid dataset.

(b) Scale-free network dataset ($|V| = 1000$, $|E| = 3994$ shown, larger graphs produced similar results.).

Figure 7: Experimental results on synthetic topologies in comparison with OBLB theoretical running time of $4W \log |E_\sigma|$. Left: OBLB Algorithm, right: Periodic algorithm. Statistical behavior over 20 executions for each parameter combination. All executions capped at 30000 rounds. That is, datapoints shown at 30000 rounds in fact indicate a running time of $\geq 30000$ rounds.

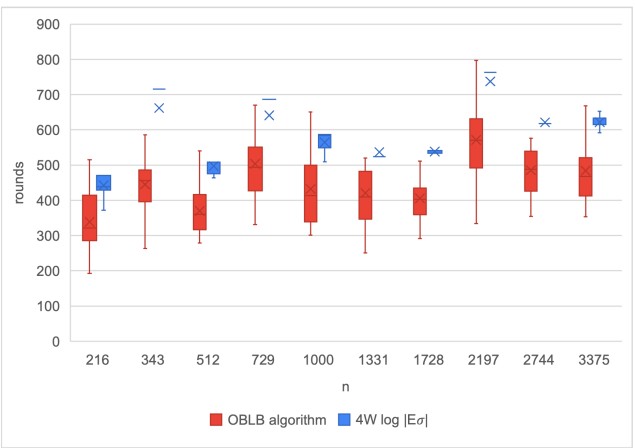

Figure 8: Experimental results on a synthetic 3D-grid topology showing that OBLB running time is independent of $n$. The OBLB theoretical running time of $4W \log |E_\sigma|$ is included to observe that the oscillation of OBLB running time is due to the oscillation of $W$ and $|E_\sigma|$. Statistical behavior over 20 executions for each parameter combination.

