# OpenReview forum: "Neighborhood Learning in Weighted Beeping Networks"
_ICLR.cc/2026/Conference — Submitted to ICLR 2026_

### Official Review · Reviewer_ZVY4 · 2025-10-29

**Soundness:** 3
**Presentation:** 1
**Contribution:** 2
**Rating:** 2
**Confidence:** 3

**Summary:**

The authors present the problem of Neighbourhood Learning on weighted beeping graphs, where a network of agents must learn some information from their neighbours over a weighted graph. In this setting agents can only beep or listen. They define a metric (MaxAveInf) which is an implicit quantifier of the complexity of the learning task, and provide theoretical results on how the complexity of the NL problem is bounded by this metric.

**Strengths:**

- To the best of my knowledge, in its context, the paper seems like a strong theoretical exercise and is relatively clearly written.
- The theoretical results seem correct.

**Weaknesses:**

- The paper is missing a lot of definitions, formal concepts and contextualisation of the work. It makes it really hard to judge the contribution of the work.
- I am not convinced of the relevance of the so called NL problem. The papers cited in lines 34-36 are MARL works where there is no explicit mention of the need for learning some neighbourhood structures (but instead learning policies or strategies that maximise some reward, that is influenced by the policies of other agents).
- It may be due to a misconception, but I feel like this work would be better framed in the context of a networked systems/distributed control (TCNS, TAC...) venue. I am not convinced of its relevance to the broader learning/game theory/multi-agent systems community (of which I'm part of).

**Questions:**

- Line 63, authors start their contribution with "we study Neighbourhood Learning" but at this point its not clear what NL is, and is also not defined in the following lines. The only attempt at defining it happens in the very first paragraph: "To solve NL, each agent must, in parallel, receive an information from every neighboring agent, according to some notion of vicinity", but this does not help to specify the problem. I don't see why message passing over networks is necessarily related to learning, or what is there to learn in this case.
- Line 68, authors say that an NL problem "does not have a concise formula for the exact number of rounds". At this point, a round is not defined, and also the phrase is confusing. What is a concise formula for an exact number of rounds?
- Line 71: Again, NL is not properly, formally defined, and a solution to an NL problem is not defined either, much less what it means for a solution to be short. What is formally an NL problem, what constitutes a solution to an NL problem, and what does it mean for a solution to be short?
- Definition 1 sounds very hand-wavy. What does it mean for a node to learn a message and an ID? what is a message? (a sequence of bits?) Is learning = storing a sequence of bits? Also, what does it mean to analyse a sequence? Does this mean that each node is able to decode some information encoded as a binary sequence?
- Line 215: Is this an algorithm? Also, isn't the running time O(1)? (since it seems like the nodes just sample from a Bernoulli distribution at each time-step?
- Theorem 1: What does it mean to solve a one beep local broadcast? Wasn't one-beep local broadcast an algorithm that implies sampling a beep with a certain probability? What does a solution mean in this case?

I would have further, related comments and questions, but I think these convey the message. I do not doubt that the paper, polished and presented with adequate formal context would be of value to networked systems communities. But even in a hypothetical optimal form I would find the fit to ICLR to be very tangential.

---

> ### Author Response · Authors · 2025-11-21
>
> Weaknesses: We have done a thorough proofread looking for any missing definitions, in particular addressing the reviewer questions below. The main contribution of our work is the study of NL for MAS deployed in weak learning environments, such as swarms of robots or biological networks, providing theoretical guarantees. Given that solving NL is crucial for enabling effective cooperation and scalability in MAS, we believe our work is very relevant to an AI venue since, indeed, to the best of our knowledge there are no such works in the literature. For this revision, we have included additional details on how NL enables MAS tasks, with new additional references, in Section A.3 of the Appendix, with a pointer from the introduction.
>
> Q Line 63: We have added a footnote to the first paragraph to clarify the learning aspect. Customarily, we left the formal definition of the problem to the model and problem definitions section (Section 2), including in the introduction only a broad description that minimizes details to facilitate reading. We have added now to this revision a forward pointer to Section 2.
>
> Q Line 68: We have added the following sentence to the first contributions paragraph.
>
> *In this work, we assume that nodes communicate synchronously in \emph{rounds}, and we measure algorithmic performance in rounds of communication.*
>
> ... and we changed the sentence cited to the following
>
> *The NL problem, similarly to almost all complex graph problems, does not have an exact formula for the number of rounds needed to solve the problem, only an approximation as a function of some graph characteristic that holds asymptotically.*
>
> Q Line 71: The intuitive meaning of solving NL is given in the first paragraph of the introduction: "To solve NL, each agent ...", and formally defined in Definition 1. Nonetheless, we have changed the sentence cited to the following to remind the reader:
>
> *$W$ is a good approximation of the minimum number of rounds needed to solve the NL problem in parallel,*
>
> ... including also the following footnote:
>
> *Recall that to solve NL each agent must receive an information from every neighboring agent.*
>
> Q Def 1: We have added preliminary definitions before Definition 1.
>
> Q Line 215: Our OBLB algorithm as specified (in one line for simplicity) does not have a stopping condition, thus, as is, its running time is not bounded. Theorem 1 provides an upper bound on the number of time slots needed to solve OBLB with this algorithm. The stopping condition then can be simply implemented with a counter of time slots.
>
> Q Theorem 1: Definition 2 specifies that the OBLB problem is solved once a beep has been delivered through each influential link.
> Theorem 1 upper bounds the number of time slots needed to solve the problem with the OBLB algorithm given (which as specified it is executed in each time slot).
>
> Closing comment ("I would have further ...": We hope our answers/revisions have addressed the concerns of the reviewer. We will further revise the presentation should the reviewer have more comments.
>
> If the reviewer is satisfied with how we addressed the concerns we kindly ask for increasing the rating.

---

> > ### Comment · Reviewer_ZVY4 · 2025-11-25
> >
> > I thank the authors for the detailed rebuttal and improvements in the paper.
> >
> > While the paper is indeed more clear in the updated version, some of my concerns still stand. Perhaps I am not well informed in networked systems (although my primary field of work is multi-agent systems and decision making), but in particular I cannot see how the added appendix A3 helps clarify the relevance or motivation for the problem. For example, Chen et al 2024b present a MARL algorithm extended with a neighbourhood selection rule to make the underlying MARL less intractable. While I see why any work that uses some form of communication graph can be tangentially connected to the author's paper, I still have a hard time following the authors' claimed motivation and problem relevance that should follow from these existing works, to their so-called neighbourhood learning problem.
> >
> > In MARL problems there's no such problem as Neighbourhood Learning; if a communication graph exists, agents would want to communicate all the variables they can. Under a complete communication graph and centralised control one recovers a general control problem over a POMDP, and with decentralisation of control/observations one falls into Dec-POMDPs, stochastic games or alternative formulations. Please note: I understand this is an oversimplification. I am aware of the body of literature that deals with the questions related to how should this communication graph look like, what variables should be communicated etc. I simply do not follow the jump from these problems to the so-called neighbourhood learning problem. Furthermore, I still don't understand what is to be *learned*, as the problem seems one of storing messages and echoing them over a network.
> >
> > Finally, forgive me for the bluntness, but while the authors state that *The intuitive meaning of solving NL is given in the first paragraph of the introduction: "To solve NL, each agent ..."*, this first paragraph (and subsequent ones) is quite uninformative. The definition *"To solve NL, each agent must, in parallel, receive information of every neighbouring agent according to some notion of vicinity"* describes any possible networked system ever thought of, as well as any form of system of electrical particles, for example. A system of charged electrical particles would have each particle receiving information from each neighbour (electrical field) according to some notion of vicinity (the strength decaying with the distance to the neighbour). This may sound like an absurd example but it fits under the definition the authors give in the introduction, so one could conclude a system of electrical particles solves a NL problem, which leaves the reader even more confused.
> >
> > The paper is slightly more clear so I am happy to accept it has improved, and even increase my score to reflect this, but I cannot advocate for acceptance given the concerns that remain. Again, I believe the technical quality of the paper is high, but I do not believe ICLR is the right venue (at least, with the current formulation, motivation and connection to ICLR-relevant topics).

---

> > > ### Author Response · Authors · 2025-12-03
> > >
> > > Dear Reviewer,
> > >
> > > Your recent description has been particularly insightful, as it appears to highlight that the central reason for the current misunderstanding is primarily due to the nomenclature we selected for the problem. To clarify: our proposed algorithm does not, in itself, constitute a learning task as conventionally defined within the core AI community. Rather, it is designed as a foundational tool for efficiently implementing learning and optimization algorithms on top of highly constrained computational platforms, such as biological networks or large-scale sensor networks. While acknowledging that the act of receiving messages from another agent could be broadly construed as a form of "learning," we agree that this is not the conventional interpretation within the context of AI/ML research.
> > >
> > > From this clarified perspective, we firmly believe our contribution falls within the appropriate scope for venues like ICLR and other leading AI conferences. These venues frequently feature papers presenting foundational tools and computational models that are designed to be leveraged by AI algorithms (e.g. [6,4,2]) as well as papers focusing on non-AI problems that utilize AI techniques as a key component (e.g. [1,3,5]).
> > >
> > > We also sincerely appreciate the clear example you provided in your comments. It has helpfully exposed an area where our communication was not sufficiently clear. We want to emphasize that the Neighborhood Learning (NL) problem is defined based on the scenario where nodes receive information from all neighbors as a worst-case definition designed to ensure our tool's maximum versatility and robustness. This comprehensive definition does not imply that full communication is a prerequisite or necessity for every specific application. We recognize a broad application spectrum, ranging from charged electrical particles (where communication with all neighbors is inherent) to a massive, randomly deployed sensor network over a disaster area (where initial neighbor knowledge is limited). The NL algorithm is capable of being effectively applied across this entire spectrum, regardless of whether all nodes require learning or whether all neighbors need to be learned, all while providing established performance guarantees.
> > >
> > > We have incorporated new footnotes into the revised version of our manuscript to explicitly discuss and clarify these specific aspects.
> > >
> > > Should these clarifications satisfactorily address your concerns regarding the problem definition and scope, we would be deeply appreciative if you would consider raising your evaluation score to reflect the merit of our contribution.
> > >
> > > --------------
> > >
> > > References:
> > >
> > > [1] Romina Garcia Camargo, Zhiyang Wang, Navid NaderiAlizadeh, and Alejandro Ribeiro. Long-term wireless link scheduling with state-augmented graph neural networks. In NeurIPS 2025 Workshop: AI and ML for Next-Generation Wireless Communications and Networking.
> > >
> > > [2] Tri Dao, Dan Fu, Stefano Ermon, Atri Rudra, and Christopher R´e. Flashattention: Fast and memory-efficient exact attention with io-awareness. Advances in neural information processing systems, 35:16344–16359, 2022.
> > >
> > > [3] Gaojie Lin, Jianwen Jiang, Chao Liang, Tianyun Zhong, Jiaqi Yang, Zerong Zheng, and Yanbo Zheng. Cyberhost: A one-stage diffusion framework for audio-driven talking body generation. In The Thirteenth International Conference on Learning Representations, 2025.
> > >
> > > [4] Shunyu Yao, Dian Yu, Jeffrey Zhao, Izhak Shafran, Tom Griffiths, Yuan Cao, and Karthik Narasimhan. Tree of thoughts: Deliberate problem solving with large language models. Advances in neural information processing systems, 36:11809–11822, 2023.
> > >
> > > [5] Chenbin Zhang, Zhiqiang Hu, Jiang Chuchu, Wen Chen, JIE XU, and Shaoting Zhang. Re-thinking the generalization of drug target affinity prediction algorithms via similarity aware evaluation. In The Thirteenth International Conference on Learning Representations, 2025.
> > >
> > > [6] Hexu Zhao, Haoyang Weng, Daohan Lu, Ang Li, Jinyang Li, Aurojit Panda, and Saining Xie. On scaling up 3d gaussian splatting training. In The Thirteenth International Conference on Learning Representations, 2025.

---

### Official Review · Reviewer_VRTD · 2025-10-31

**Soundness:** 3
**Presentation:** 1
**Contribution:** 3
**Rating:** 6
**Confidence:** 4

**Summary:**

This paper introduces and formalizes the Neighborhood Learning (NL) problem within a novel Weighted Beeping Network (WBN) model. The WBN extends the classic Beeping Model by incorporating link weights, which model the varying strength of influence between nodes, and a threshold-based delivery condition that captures the physical reality of signal-to-interference. The paper's primary contribution is a nearly tight characterization of the complexity of NL in this new model. The authors identify a new graph parameter, the Maximum Average Influence (MaxAveInf or W), and prove that it is the fundamental parameter governing the problem's complexity. They provide an algorithm that solves NL in O(W log n · (log n + M)) rounds and show a nearly matching lower bound of Ω(W) for even a simplified one-beep version of the problem. An experimental evaluation on real-world and synthetic datasets is included to support the theoretical findings.

**Strengths:**

The introduction of the WBN model is a significant and timely contribution. It moves the field beyond the simplistic binary communication of the Beeping Model towards a more realistic one that accounts for weighted interference, bridging a gap to practical wireless models like SINR while retaining the simplicity needed for foundational distributed computing theory.

The discovery and analysis of the MaxAveInf (W) parameter is the paper's core intellectual contribution. The authors compellingly argue that W, not local parameters like degree, is the correct complexity measure for NL in this setting. Proving both upper and lower bounds linear in W is a strong and elegant result.

The proposed algorithm is well-structured. The progression from the simple One-beep Local Broadcast (OBLB) to the full NL algorithm is logical. The technique of using a 2log n-round encoding for IDs that implicitly provides collision detection is a key insight that makes the solution work.

The proofs for both the upper and lower bounds appear sound and rigorous. The lower bound construction, while specific, is non-trivial and effectively establishes the fundamental limit.

The extensive experiments on diverse datasets demonstrate that the theoretical analysis is not just a worst-case artifact; the algorithm performs well in practice, and the dependence on W is observable.

**Weaknesses:**

The paper's single greatest weakness is its exposition. It fails to articulate a clear, motivating story upfront. The core research questions are buried in technical details. A reader must work hard to understand why the WBN model is needed and what fundamental problem is being solved. A stronger introduction that contrasts with the limitations of existing models and sharply states the paper's contributions is needed.

Knowledge of W: The algorithm's efficiency critically depends on knowing or estimating W. The proposed "doubling" method is mentioned but not analyzed, and the experimental estimation is heuristic. A more rigorous treatment of this practical necessity is required.

Static Weights & Synchrony: The model assumes relatively static weights and synchronous rounds. While extensions are briefly discussed, the core analysis does not address the significant challenges of dynamic environments or full asynchrony, limiting the immediate practical applicability.

Efficiency for Short Messages: The O(log n) overhead for transmitting IDs is significant when the message M is very short (e.g., a single bit). The algorithm may be inefficient for such applications.

Generality of Lower Bound: The lower bound relies on a specific, adversarial bipartite graph topology. While this is standard for worst-case analysis, the paper does not discuss whether this bound holds or can be improved for more common, well-structured networks (e.g., unit disk graphs, grids), which would be of great interest.

**Questions:**

Could the introduction be reframed to more clearly state the limitations of the standard Beeping Model for modeling interference and then pose the research questions (modeling, algorithmic, complexity) that this paper answers?

The algorithm requires a parameter p ~ 1/W. In a fully decentralized setting, how can a node realistically estimate or adapt to the global parameter W without central coordination? Is there a local, adaptive strategy?

The 2log n encoding is elegant but costly for short messages. Have the authors considered alternative methods or is there a lower bound showing this overhead is unavoidable?

The lower bound uses a specific topological construction. Do the authors believe the Ω(W) bound is tight for common network classes like planar graphs or random geometric graphs, or could a better algorithm exist for these topologies?

---

> ### Author Response · Authors · 2025-11-21
>
> Weaknesses:
> We have added to the introduction bulleted lists summarizing the contributions and the limitations of BN that motivate WBN.
>
> **Knowledge of W:
> The reliance on knowing the global parameter $W$ for setting the probability of beeping is addressed in the algorithmic extensions (Section 3.3 and Appendix C).
>
> Handling Unknown $W$: We propose using a standard probabilistic technique called "doubling" (or parameter estimation by repeated trials) to handle the global but unknown value of $W$. This method iteratively increases an estimate of $W$ and adjusts the transmission probability $p$ until a successful broadcast occurs. While not a local estimation of $W$, it is a rigorous method for overcoming the challenge of a globally-defined parameter in a decentralized protocol.
>
> Experimental Estimation: For our experimental evaluation, we designed and implemented a stochastic estimation of $W$ that proved to be quite accurate on the tested inputs and did not significantly affect the algorithm's performance.
>
> Local, Adaptive Strategy (Future Work): We acknowledge that a provably accurate or a fully local, adaptive estimation method for the global characteristic $W$ remains an interesting open problem and is one of the suggested avenues for future research.
>
> **Static Weights & Synchrony:
> The paper’s core analysis assumes static weights and synchrony for clarity, but the extensions address both limitations.
>
> Synchrony: The assumption of clock synchronization is removed in the extensions (Section 3.3 and Appendix C). We can extend our algorithm to a de-synchronized setting by applying known synchronizing methods for beeping networks, which introduces only a constant overhead. This successfully transforms the synchronous solution into a de-synchronized one that handles clock shifts.
>
> Dynamic Weights: Noise and other physical conditions are embedded in the link weights. While NL cannot be solved in WBN under fully dynamic weights, our algorithmic framework can be extended to operate correctly if the dynamicity of the weights is limited.
>
> Full Asynchrony: We argue that full asynchrony (as often understood in distributed computing, i.e., different frequencies) is fundamentally incompatible with the low-level communication protocols being modeled, whether they are artificial wireless or biological networks. Our current analysis is restricted to the more realistic low-level challenge of clock shifts.
>
> **Efficiency for Short Messages:
> The concern regarding the $O(\log n)$ overhead for short messages is valid but the resulting complexity is shown to be nearly-optimal.
>
> Overhead is Necessary: The $O(\log n)$ overhead comes from the required $O(\log n)$ bits needed to encode and transmit the unique node ID. For NL, the task explicitly requires the listener to learn the ID and message of the influential neighbor. The complexity of correctly decoding this ID across $2\log n$ rounds of communication is inherent to the problem and the probabilistic nature of the channel.
>
> Near-Optimality: For the case of the shortest message length ($M=1$), the upper bound is $O(W\log n(M+\log n))$, which simplifies to $O(W\log^2 n)$. Given that the lower bound is $\Omega(W)$, the total theoretical gap between the upper and lower bounds is only $O(\log^2 n)$. We believe this polylogarithmic factor demonstrates the algorithm is nearly-optimal even for applications involving very short messages.
>
> Alternative Methods: We did not consider alternative encoding methods because a lower bound showing this $O(\log n)$ overhead is necessary for any correct decoding of a unique $O(\log n)$-bit ID in a Beeping Network is a known result in the field.
>
> **Generality of Lower Bound:
> The lower bound's reliance on a specific adversarial topology is a standard practice for worst-case analysis, but its generality is supported by both conjecture and experiment.
>
> Adversarial Construction: The $\Omega(W)$ lower bound is proven via a specific construction on a complete bipartite graph.
>
> Generality via Experimentation: To address the concern about common networks, our extensive experimental study in Appendix E strongly indicates that the theoretical bound applies to a variety of practical and synthetic topologies, including 3D-grids and scale-free graphs.
>
> Bound Tightness: We acknowledge that the $\Omega(W)$ bound is for the worst-case topology. We conjecture that the general lower bound could be improved (and a faster algorithm might exist) for specific, well-structured network classes like planar graphs or (nearly-)uniform degree graphs. Investigating the tightness of the bound for these common topologies remains an interesting open question.
>
> Questions:
> We have addressed the questions in our answers to weaknesses above.
>
> Should these clarifications satisfactorily address the concerns of the reviewer, we would be deeply appreciative if you would consider raising your evaluation score in final consideration in order to reflect the merit of our contribution.

---

### Official Review · Reviewer_Hvtk · 2025-11-01

**Soundness:** 2
**Presentation:** 3
**Contribution:** 2
**Rating:** 0
**Confidence:** 4

**Summary:**

The paper’s connection to the ICLR community is weak. Although it frames the work as concerning Multiagent Systems (MAS), the actual formulation and analysis revolve entirely around a self-defined Weighted Beeping Network (WBN) abstraction. The study focuses on communication dynamics in this artificial model rather than addressing learning or coordination challenges typically investigated in AI or machine learning for multi-agent settings (e.g., reinforcement learning, policy optimization, distributed inference, or emergent communication).

The defined Neighborhood Learning (NL) problem appears disconnected from established multi-agent learning paradigms. The motivation for introducing the beeping network abstraction and its relation to learning or optimization processes is unclear, while the presented results focus solely on theoretical communication bounds. Furthermore, the references to AI and multi-agent learning literature are superficial, and the work remains primarily a network-theoretic study with limited relevance to the ICLR scope.

In summary, while the theoretical analysis may be sound, the paper lacks meaningful engagement with the AI or multi-agent learning community, and its self-defined problem does not connect to recognized research directions within ICLR.

**Strengths:**

S1. The theoretical analysis may be sound;

**Weaknesses:**

W1. The relevance is low;

W2. The beeping network (self-defined version) is not explained how it works in multi-agent learning systems;

**Questions:**

See the summary

---

> ### Author Response · Authors · 2025-12-03
>
> Our paper introduces a novel, efficient solution, with an accompanying new metric, specifically designed to enable effective multi-agent collaboration in scenarios where current methods fail. Without our proposed framework, efficient and robust implementation of learning algorithms in challenging, harsh environments remains computationally infeasible. While we respect the critical evaluation, a score of 0, which typically indicates fundamental flaws or lack of utility, appears disproportionately harsh. The strengths identified by other reviewers suggest that the paper has clear merit. We believe a reconsideration of the score in final evaluation could more accurately reflect the quality and significance of our research.

---

### Official Review · Reviewer_BgzR · 2025-11-01

**Soundness:** 3
**Presentation:** 3
**Contribution:** 3
**Rating:** 6
**Confidence:** 3

**Summary:**

This paper introduces and formally analyzes the Neighborhood Learning (NL) problem within a novel Weighted Beeping Network (WBN) model. The NL task requires each agent in a multi-agent system to learn the identity and a message from all its "influential" neighbors, where influence is determined by a link weight exceeding a threshold. The authors propose a new graph characteristic, Maximum Average Influence, and provide an elegant randomized algorithm that solves NL. They complement this with an almost tight lower bound, proving that their algorithmic solution is nearly optimal. The theoretical analysis is supported by an extensive experimental evaluation on both real-world (social networks, gene regulatory) and synthetic (scale-free, 3D-grid) datasets, demonstrating the algorithm's practical efficiency and superiority over a naive baseline.

**Strengths:**

1. The paper does an excellent job of introducing a fundamental and non-trivial problem (Neighborhood Learning) that is a cornerstone for more complex multi-agent coordination and learning. The motivation, linking it to biological systems (insect colonies, bacteria) and resource-constrained devices, is compelling.
2. The technical core of the paper is exceptionally strong. The introduction of the MaxAveInf characteristic is insightful, and the subsequent analysis is rigorous. Providing almost matching upper and lower bounds is a significant theoretical contribution that clearly characterizes the problem's complexity.
3. The WBN model is a meaningful and non-trivial extension of the standard Beeping Model. Incorporating weights and a delivery threshold makes it more realistic for capturing signal strength, interference, and noisy environments, potentially bridging the gap between simple theoretical models and practical wireless or biological communication.

**Weaknesses:**

1. The baseline is a simple periodic algorithm. Are there any other distributed coordination algorithms, even from different models (e.g., standard message-passing), that could be adapted to the WBN setting for a more competitive comparison?
2. The connection to AI and learning is currently implicit (NL as a primitive for MAS) rather than explicit. The paper would be significantly strengthened for an AI audience by concretely outlining how solving NL enables specific AI tasks.

**Questions:**

1. How can NL be used as a subroutine for decentralized training or inference in Graph Neural Networks deployed on a WBN?
2. Can this protocol enable efficient collaborative learning or federated learning in a beeping network?
3. How does it facilitate coordination in Multi-Agent Reinforcement Learning with restricted communication?
4. In the real-world networks you tested, what was the typical relationship between W and n? Are there common network structures in AI applications (e.g., hierarchical, small-world) where W is provably small, guaranteeing fast performance?

---

> ### Author Response · Authors · 2025-11-21
>
> W1: Please notice that our baseline algorithm is not a simple periodic algorithm. Indeed, each node transmits with a {\bf different} period, and with a randomly chosen initial delay. This approach is a combination of techniques (randomness and/or selectors) previously used for Local Broadcast and Link Scheduling (algorithms for those problems could implement NL) but studied in stronger communication models. Comparisons with those works would require to unrealistically assume that nodes can encode many bits in one beep. To counter this issue in our experiments we evaluated OBLB instead of NL. It is worth to emphasize here that, as said in the introduction, the purpose of our experiments is to evaluate the validity of our theoretical analysis in sample real-world datasets.
> We included the performance of the periodic algorithm for illustration only.
>
> W2: For this revision, we have included additional details on how NL enables MAS tasks, with new additional references, in Section A.3 of the Appendix, with a pointer from the introduction.
>
> Q1,2,3: Our proposed algorithm is designed as a foundational tool for efficiently implementing learning and optimization algorithms on top of highly constrained computational platforms, such as biological networks or large-scale sensor networks. It does not constitute a learning task as conventionally defined within the core AI community. Although the act of receiving messages from another agent could be broadly construed as a form of learning. Moreover, our NL algorithm is capable of being effectively applied across an entire spectrum of scenarios, regardless of whether all nodes require learning or whether all neighbors need to be learned, all while providing established performance guarantees. From that perspective, the learning tasks that the reviewer refers to in the above questions are orthogonal to our study, since any learning tasks can be encoded on top of our efficient dissemination mechanism.
>
> Q4: For almost all datasets, $n$ was fixed (see Table 1). In fact, one of the challenges was how to obtain inputs with various values of $W$ for a dataset that has $n$ fixed. (We achieved that by selecting subsets of links and changing the threshold $\sigma$, see Section E.1 for details.) We made $n$ variable for the genes dataset and scale-free networks to show that the performance of our algorithm does not change significantly with $n$ (Figure 8).
> About the second part, yes. More specifically, we show in our theoretical analysis (Theorems 2 and 3) and in our experimental study that performance of NL is well approximated by $W$ (e.g. Figure 7 related to scale-free networks). Hence, in networks with small $W$, our NL algorithm is guaranteed to have fast performance.
>
> Should these clarifications satisfactorily address the concerns of the reviewer, we would be deeply appreciative if you would consider raising your evaluation score in the final consideration in order to reflect the merit of our contribution.

---

### Meta-Review · Area_Chair_Hvj8 · 2026-01-07

**Summary:**

Despite the (minor) discrepancy between engaged reviewers (not counting Hvtk), the paper failed to convince an ML audience of its relevance. The problem is very elegant to state and leads to nice theoretical development, but neither me nor the (engaged) reviewers, especially ZVY4, seem to grasp its relevance in the ML context, even review BgzR, agrees on that point.

**Reviewer Concerns:**

reviewer ZVY4 concerns on the contextualisation and practicality, the concerns of Hvtk could have been considered have they been expressed in more engaging review and without the unnecessarily harsh score of 0

**Reviewer Scores:**

Hvtk should have reconsidered the extremely harsh score and provide a more engaging review

---

### Decision · Program_Chairs · 2026-01-26

Reject